# FLASHFFTCONV: EFFICIENT CONVOLUTIONS FOR LONG SEQUENCES WITH TENSOR CORES

**Daniel Y. Fu**[∗], **Hermann Kumbong**[∗], **Eric Nguyen, Christopher Ré**
[∗]Equal contribution. Stanford University.
{danfu,kumboh,etnguyen,chrismre}@stanford.edu

## ABSTRACT

Convolution models with long filters have demonstrated state-of-the-art reasoning abilities in many long-sequence tasks but lag behind the most optimized Transformers in wall-clock time. A major bottleneck is the Fast Fourier Transform (FFT)—which allows long convolutions to run in $O(N \log N)$ time in sequence length $N$ but has poor hardware utilization. In this paper, we study how to optimize the FFT convolution. We find two key bottlenecks: the FFT does not effectively use specialized matrix multiply units, and it incurs expensive I/O between layers of the memory hierarchy. In response, we propose FLASHFFTCONV. FLASHFFTCONV uses a matrix decomposition that computes the FFT using matrix multiply units and enables kernel fusion for long sequences, reducing I/O. We also present two sparse convolution algorithms—1) partial convolutions and 2) frequency-sparse convolutions—which can be implemented simply by skipping blocks in the matrix decomposition, enabling further opportunities for memory and compute savings. FLASHFFTCONV speeds up exact FFT convolutions by up to 7.93× over PyTorch and achieves up to 4.4× speedup end-to-end. Given the same compute budget, FLASHFFTCONV allows Hyena-GPT-s to achieve 2.3 points better perplexity on the PILE and M2-BERT-base to achieve 3.3 points higher GLUE score—matching models with twice the parameter count. FLASHFFTCONV also achieves 96.1% accuracy on Path-512, a high-resolution vision task where no model had previously achieved better than 50%. Furthermore, partial convolutions enable longer-sequence models—yielding the first DNA model that can process the longest human genes (2.3M base pairs)—and frequency-sparse convolutions speed up pretrained models while maintaining or improving model quality.

## 1 INTRODUCTION

A key challenge in machine learning is to efficiently reason over long sequences. Recently, convolutions have emerged as a key primitive for sequence modeling, underpinning state-of-the-art performance in language modeling (Ma et al., 2022; Poli et al., 2023; Wang et al., 2022a; Fu et al., 2023a), time-series analysis (Tang et al., 2022; Zhang et al., 2022b; Gu et al., 2021; Fathullah et al., 2023), computer vision (Wang et al., 2023; Liu et al., 2022; Nguyen et al., 2022), DNA modeling (Nguyen et al., 2023), and more (Li et al., 2022; Islam et al., 2023; Kim & Park, 2023; David et al., 2022; Miyazaki et al., 2023; Mehari & Strodthoff, 2023). Despite these strong quality results—and other benefits ranging from better scaling in sequence length (Gu et al., 2021) to greater stability (Tay et al., 2021a; Bietti & Mairal, 2017)—convolutional sequence models still lag behind Transformers in wall-clock time.

A major reason is poor hardware support. Unlike classical convolutions used in vision applications, which often have short filters (e.g., $3 \times 3$ or $7 \times 7$ (Krizhevsky et al., 2012; He et al., 2016)), convolutions for sequence modeling often use filters as long as the input sequence (Romero et al., 2021b; Li et al., 2022). Such long filters necessitate the use of the FFT convolution algorithm, which computes the convolution between an input $u$ and convolution kernel $k$ via a conversion to frequency space:

$$(u*k)[i] = \sum_j^i u[i]k[j-i] \quad \cong \quad u*k = \mathcal{F}^{-1}(\mathcal{F}u \odot \mathcal{F}k), \tag{1}$$

where $\mathcal{F}$ is the FFT, which can be computed in $O(N \log N)$ time in sequence length $N$, and $\odot$ is elementwise multiplication. Despite its asymptotic efficiency, the FFT convolution algorithm has

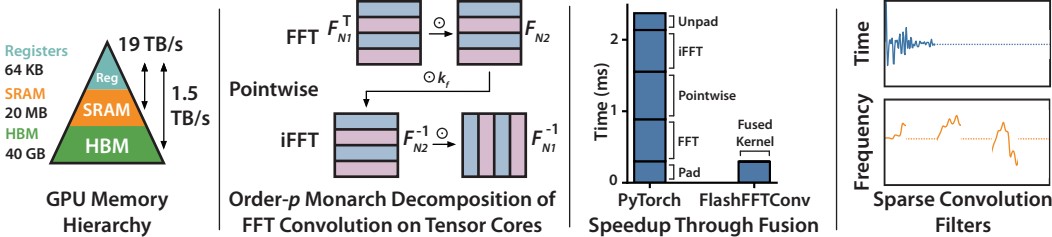

Figure 1: **Left:** GPU memory hierarchy. **Middle left:** Order-$p$ Monarch decomposition of FFT convolution, with $p=2$. **Middle right:** Kernel fusion for end-to-end speedup. **Right:** FLASHFFTCONV introduces analogues of sparsity for convolutions.

poor wall-clock time on modern accelerators. In contrast, systems advances have pushed Transformers to the limits of modern accelerators—achieving more than 72% FLOP utilization end-to-end with FlashAttention-v2 (Dao et al., 2022b; Dao, 2023).

In this paper, we study how to optimize the FFT convolution algorithm on modern accelerators, to enable longer-context abilities. Just as systems advances such as FlashAttention yielded improvements in modeling quality (Ahdritz et al., 2022; Li et al., 2023) and the development of new attention algorithms (Paliotta et al., 2023; Athiwaratkun et al., 2023; Kwon et al., 2023; Liu & Abbeel, 2023), we hope that understanding how to optimize the FFT convolution can also inspire algorithmic innovation, thus improving the quality of convolutional sequence models.

For short sequences, the FFT convolution is relatively easy to optimize. Kernel filters are often shared across many batches, which allows pre-computing the FFT of the filter $k_f = \mathcal{F}k$ and re-using it in a batch: $(u*k) = \mathcal{F}^{-1}(\mathcal{F}u \odot k_f)$. Thus the FFT convolution is pleasantly parallel across batches and filters, and intermediate outputs of the convolution can be cached in SRAM or registers via kernel fusion.

However, as sequence length increases, we find that two key bottlenecks emerge. First, FFT convolutions do not effectively use the specialized matrix-matrix multiply units available on modern accelerators—e.g., the H100 can use tensor cores to compute matrix-matrix multiply at 1.0 PetaFLOP/s compared to 67 TeraFLOP/s for general arithmetic. Second, sequences become too large to fit in SRAM, and kernel fusion fails, resulting in expensive I/O costs (Figure 1 middle right). These I/O costs can be exacerbated by padding operations for causality, and conversions from real-valued inputs/outputs to complex-valued FFT intermediates.

In response, we propose FLASHFFTCONV, a new system that optimizes the FFT convolution for long sequences using a Monarch decomposition of the FFT. An order-$p$ Monarch decomposition rewrites the FFT as a series of $p$ matrix-matrix multiply operations (Figure 1 middle left), which can be efficiently mapped onto hardware (Dao et al., 2022a). The order $p$ controls the number of matrix multiply operations and introduces a tradeoff: higher values of $p$ incur lower FLOP cost via smaller matrices, but require more I/O to communicate intermediate results. Using a simple GPU cost model, we show how to adjust $p$ based on the sequence length to balance the FLOP cost and I/O cost. This decomposition introduces a second benefit: a reduction in the amount of the sequence that needs to be kept in SRAM, which makes kernel fusion viable at longer sequence lengths. As a result, FLASHFFTCONV scales across four orders of magnitude in sequence length, from 256 to 4 million. FLASHFFTCONV also exploits a real-valued FFT algorithm to cut the length of the FFT operation in half (Sorensen et al., 1987), and selectively skips portions of the matrix-multiply operations when the input is zero-padded.

Finally, the matrix view of the FFT convolution presents a natural interface to implement two architectural modifications: partial convolutions, which learn with $k$ that is shorter than the input sequence, and frequency-sparse convolutions, which zero out portions of the kernel $k_f$ in frequency space. These can be viewed as convolutional analogues to sparse/approximate attention in Transformers (Han et al., 2015a;b; Kitaev et al., 2020; Paliotta et al., 2023; Beltagy et al., 2020), and map naturally on to FLASHFFTCONV: both algorithms can be implemented simply by skipping portions of the matrix decomposition, thus reducing memory footprint and wall-clock runtime.

**Evaluation** We show that FLASHFFTCONV speeds up the FFT convolution, yielding higher-quality, more efficient, and longer-sequence models.

- **Quality** FLASHFFTCONV improves the quality of convolutional sequence models via better efficiency: for the same compute budget, FLASHFFTCONV allows Hyena-GPT-s to achieve 2.3 points better perplexity (Poli et al., 2023), and allows M2-BERT-base (Fu et al., 2023a) to achieve up to 3.3 higher average GLUE score—a gain in performance equivalent to doubling the parameters of the model.
- **Efficiency** FLASHFFTCONV makes convolutions more efficient across four orders of magnitude in sequence length, yielding speedups of up to $7.93\times$ and memory savings of up to $5.60\times$ over PyTorch. FLASHFFTCONV achieves up to 62.3% end-to-end FLOP utilization—only 10% less than FlashAttention-v2—and is faster in wall-clock time than FlashAttention-v2 end-to-end at sequence lengths 2K and longer due to lower FLOP costs.
- **Longer Sequence Models** FLASHFFTCONV enables longer-sequence models. In high-resolution image classification, FLASHFFTCONV yields the first model that can solve the challenging Path-512 task (sequence length 256K) from the long range arena benchmark (Tay et al., 2020). In DNA modeling, FLASHFFTCONV uses partial convolutions to extend HyenaDNA (Nguyen et al., 2023) to 4M sequence length—yielding the first model that can embed the longest human genes (up to 2.3M base pairs) at single nucleotide resolution.

Overall, we hope that FLASHFFTCONV enables further adoption of convolutional sequence models and that the insights from our work helps inform the design of better hardware-efficient architectures.

## 2 BACKGROUND

We provide some background on the FFT convolution and the Monarch FFT decomposition, and discuss the performance characteristics of GPUs.

### 2.1 FFT CONVOLUTION

Recall the definition of a convolution operation: $(u * k)[i] = \sum_j^i u_j k_{i-j}$. Computing this formula directly incurs $O(NN_k)$ FLOPs in sequence length $N$ and kernel length $N_k$. For long convolutions, where $N_k = N$, a popular strategy is to use the Fourier transform to convert the signal $u$ and kernel $k$ to the frequency domain, and compute the convolution using pointwise multiplication in frequency domain, using Equation 1. Critically, a Fourier transform $\mathcal{F}_N$ over an input of length $N$ can be computed in $O(N\log N)$ time using the FFT—bringing the overall cost of the long convolution from $O(N^2)$ to $O(N\log N)$. We note that the FFT convolution technically computes a circular convolution $\sum_j^N u_j k_{i-j}$, where $i-j<0$ loops back to the end of $k$. For this reason, $u$ and $k$ are often padded with zeros to compute a causal convolution.

**Monarch FFT Decomposition** For $N = N_1 N_2$, an order-2 Monarch FFT decomposition rewrites $\mathcal{F}_N = \mathbf{P}(\mathbf{I}_{N_2} \otimes \mathcal{F}_{N_1})\mathbf{D}\mathbf{P}^{-1}(\mathbf{I}_{N_1} \otimes \mathcal{F}_{N_2})\mathbf{P}$, where $\otimes$ denotes the Kronecker product, $\mathcal{F}_N$ is the $N \times N$ discrete Fourier matrix, $\mathbf{P}$ is a permutation matrix that reshapes the input to $N_1 \times N_2$, transposes it to $N_2 \times N_1$, and then reshapes it back to $N$, and $\mathbf{D} \in \mathbb{C}^{N \times N}$ is a diagonal matrix containing correctional values called Twiddle factors (Bailey, 1989). Higher-order Monarch decompositions recursively apply the order-2 decomposition to $\mathcal{F}_{N_1}$ or $\mathcal{F}_{N_2}$, which reduces FLOP costs but increases the number of permutation operations, increasing I/O cost.

### 2.2 GPU PERFORMANCE CHARACTERISTICS

We provide some background on the GPU memory hierarchy and available compute units, as well as compute-bound vs. memory-bound operations. We focus on GPU programming in this paper, but the general principles extend to most modern hardware accelerators (Jouppi et al., 2023; Zhang et al., 2022a; Lavely, 2022; Emani et al., 2021).

**GPU Compute Model and Memory Hierarchy** GPUs have a memory hierarchy consisting of high-bandwidth memory (HBM), shared memory (SRAM), and registers, as shown in Figure 1 Left. Lower/larger levels of the memory hierarchy have more space but are much slower, whereas higher/smaller levels of the memory hierarchy have less space but are much faster (NVIDIA, 2017; 2020; 2022). The memory hierarchy is closely tied to the GPU compute model. A GPU is composed of many independent streaming multiprocessors (SMs), each of which is composed of independent threads. HBM is shared among all SMs, but each SM has an independent SRAM. The SRAM is shared among all the threads in the SM. Each thread has access to its own registers, but cannot access the

registers of other threads. Thus, performing global operations between SMs requires moving data to and from HBM, whereas independent work in each SM can remain local to SRAM.

**GPU Compute Units** Modern GPUs (since the V100 (NVIDIA, 2017)) have specialized matrix multiply units called tensor cores, which can compute matrix-matrix multiply operations with much higher TFLOPs than the general-purpose compute units. For example, the H100 tensor core can compute matrix multiplication between $16 \times 16$ matrices at 1.0 PFLOPs, whereas the general-purpose compute units can only compute at 67 TFLOPs (NVIDIA, 2022).

**Memory-Bound vs. Compute-Bound Operations** GPU operations can be memory-bound or compute-bound. Memory-bound operations are bottlenecked by the amount of I/O between HBM and registers they need to perform, and are limited by the bandwidth of the memory hierarchy. Examples include simple pointwise operations such as addition or multiplication, as well as most traditional FFT implementations. Compute-bound operations are bottlenecked by the amount of FLOPs they need to execute, and are limited by the speed of the compute units. Examples include large matrix multiply operations.

**Kernel Fusion** A popular method for reducing I/O costs is kernel fusion—loading data for multiple operations into SRAM, computing them independently in each SM, and then writing the final results back to HBM. Kernel fusion is common (and can be automated) for pointwise operations (Paszke et al., 2019), but is more challenging for complex operations that require referencing multiple pieces of data. For example, fusing the operations in attention was not common until the development of FlashAttention (Dao et al., 2022b).

## 3 FLASHFFTCONV

Section 3.1 provides a broad overview of FLASHFFTCONV and shows how to adapt the Monarch FFT decomposition to convolutions, which involves broadcasting the matrix multiply in parallel across the input sequence. We also describe our kernel fusion strategy and how we exploit domain-specific properties of the convolution in ML for further optimization. Section 3.2 presents a cost model characterizing the relative cost of different order-$p$ decompositions of the FFT as sequence length changes, along with a simple heuristic for selecting $p$ given hardware characteristics. Finally, Section 3.3 discusses architectural extensions by presenting analogues to sparsity in convolutional kernels.

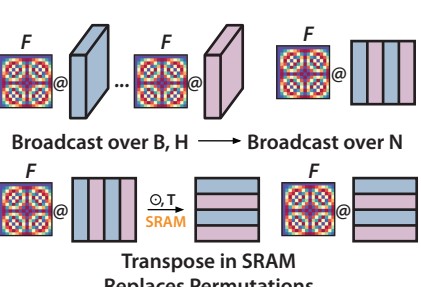

**Broadcast over B, H** ⟶ **Broadcast over N**

**Transpose in SRAM Replaces Permutations**

Figure 2: **Top:** FLASHFFTCONV adapts the Monarch FFT decomposition to broadcast matrix multiply operations over the sequence instead of over the batch and hidden dimensions. **Bottom:** This converts HBM permutations into simple matrix transpose operations in SRAM.

### 3.1 FLASHFFTCONV ALGORITHM

We describe the core FLASHFFTCONV algorithm. Algorithm 1 provides an overview. We first describe how we adapt the Monarch FFT decomposition for convolutions. Then, we discuss how the Monarch decomposition enables kernel fusion for long sequences. We conclude by presenting domain-specific optimizations.

**Adapting Monarch for Fusion** The Monarch FFT decomposition, as well as classical algorithms such as Bailey's FFT algorithm (Bailey, 1989), traditionally broadcasts the matrix operation against the batch dimension and the hidden dimension (Figure 2 top left). This allows each $\mathcal{F}_{N_1}$ operation in the $\mathbf{I}_{N_2} \otimes \mathcal{F}_{N_1}$ matrix to run independently. However, it also makes kernel fusion difficult; fusing across the matrix multiply and permutation operations requires loading at least 16 sequences at once into SRAM to fill out the matrix multiply unit—limiting sequence length to around $2K$ on A100 and H100.

Instead, we broadcast the matrix operation across the entire sequence (Figure 2 top right) and run the algorithm in parallel across the batch and hidden dimensions. This reduces the SRAM requirements for kernel fusion, since we only need to load a single sequence into SRAM at a time—allowing us to fuse the entire kernel for sequences up to 32K in fp16/bf16 on A100 and H100. Broadcasting along the sequence has an added benefit: the permutations simply become matrix transposes (Figure 2 bottom), which can be done quickly using well-established routines on-chip (NVIDIA, 2020). The core algorithm is shown in Algorithm 1 for a two-way decomposition. Higher-order decompositions and more details are given in Appendix B.

---

**Algorithm 1** FLASHFFTCONV core algorithm, with order-2 Monarch decomposition. We assume $N = N_1^2$ for simplicity here.

---

**Input:** Input $u \in \mathbb{R}^{B \times H \times N}$, convolution kernel $k_f \in \mathbb{C}^{H \times N}$, FFT matrices $\mathbf{F} \in \mathbb{C}^{N_1 \times N_1}$, $\mathbf{F}^{-1} \in \mathbb{C}^{N_1 \times N_1}$, Twiddle factors $t \in \mathbb{C}^N$, $t_{inv} \in \mathbb{C}^N$, $B$ tile size $B_{tile}$, $H$ tile size $H_{tile}$.
**Output:** Output $y \in \mathbb{R}^{B \times H \times N}$.
 **for** SMs in parallel across $B/B_{tile} \times H/H_{tile}$ **do**
  Load $\mathbf{F}, \mathbf{F}^{-1}, t, t_{inv}$ from HBM.
  **for** $h \leftarrow 1$ to $H_{tile}$ **do**
   Load $\mathbf{K_f} \leftarrow k_f[h]$ from HBM, reshaped to $N_1 \times N_1$.
   **for** $b \leftarrow 1$ to $B_{tile}$ **do**
    Load $\mathbf{X} \leftarrow u[b,h]$ from HBM, reshaped to $N_1 \times N_1$.
    $\mathbf{X} \leftarrow ((\mathbf{F}^\top \mathbf{X}) * t)\mathbf{F}$        ▷ FFT, decomposed into two steps
    $\mathbf{X} \leftarrow \mathbf{X} * \mathbf{K_f}^\top$         ▷ Elementwise multiply with $k_f$
    $\mathbf{Y} \leftarrow ((\mathbf{X}\mathbf{F}^{-1})^\top * t_{inv})\mathbf{F}^{-1}$   ▷ Inverse FFT, decomposed into two steps
    Write $\mathbf{Y}^\top$ to HBM.

---

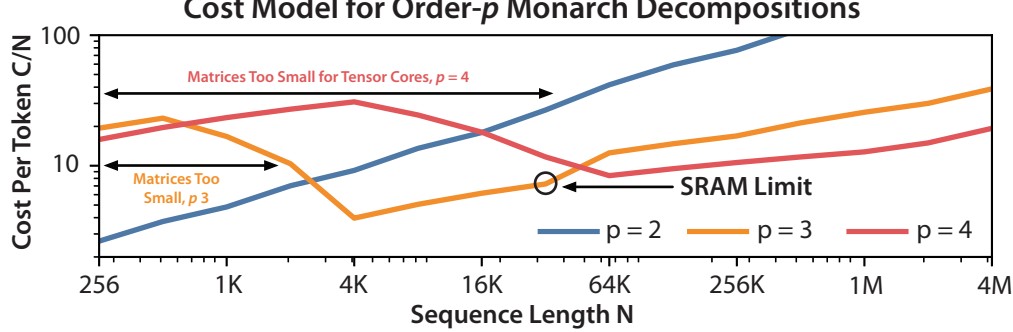

Figure 3: Compute costs of different order-$p$ Monarch decompositions as sequence length increases on A100. For short sequences, higher-order decompositions break the FFT down into matrices that are too small for tensor cores. At length 32K, the sequence is too long to fit into SRAM, and higher-order decompositions are necessary for fusion.

**Kernel Fusion and Recomputation** The Monarch decomposition allows kernel fusion for long sequences. Inner layers of the decomposition do not require the entire sequence, which reduces the SRAM requirements for fusion. Thus, for long sequences, we can fuse the innermost matrix operations and elementwise multiplications, and take an I/O each for the outermost matrix operations. We use also use **recomputation** in the backward pass to reduce the memory footprint and I/O cost. Instead of storing intermediate results on HBM for the backward pass (e.g., the intermediate result of $\mathcal{F}_N u$), we simply recompute them in the backward pass.

**Domain-Specific Optimizations** Finally, we use a few domain-specific optimizations to adapt the convolution specifically for the sequence learning workload. First, since the convolutions used in sequence learning are real-to-real convolutions (with real kernel weights), we can use a classic algorithm called one-stage decimation in time to compute the FFT of a sequence of length $N$ using a complex FFT of length $N/2$ (see Appendix B)—cutting the FFT cost in half. Second, inputs and outputs are often padded with zeros in the convolution to compute a causal convolution (Gu et al., 2021; Poli et al., 2023; Fu et al., 2023a). We special-case this padding, and use it to eliminate half of the outermost matrix multiply operations in the FFT and iFFT. We also fuse in additional operations around the convolution, such as elementwise-gating, to further reduce I/O.

### 3.2 COST MODEL OF ORDER-$p$ MONARCH DECOMPOSITION

We present a formal cost model for an order-$p$ Monarch decomposition of the convolution based on sequence length. The cost model accounts for both the cost of compute and I/O, similar to a roofline analysis (Hennessy & Patterson, 2011). Let $B$ and $H$ be the batch size and model hidden dimension, respectively, and assume that we compute the convolution in half precision. Let $N$ be the sequence

length, and let $N = \Pi_{i=1}^{p} N_i$ be the product of $p$ factors. For simplicity, we will assume that $N$ is a power of 2. Let $\mu$ be the size of the matrix-matrix multiply unit on the GPU (e.g., 16 for A100 (NVIDIA, 2020) and H100 (NVIDIA, 2022)). Let $\tau_G$ and $\tau_M$ be the empirically-achievable FLOPs on the GPU for general-purpose arithmetic, and matrix-matrix multiply arithmetic, respectively. For convenience, define $\gamma(N_i)$ as a helper function that returns $\tau_G$ if $N_i < \mu$, and $\tau_M$ if $N_i \geq \mu$. Finally, let $\sigma_H$ and $\sigma_S$ be empirically-achievable bandwidth for HBM and SRAM, respectively. Sample values for these constants are given in Appendix D.

Now, we can present the cost of an FFT convolution with an order-$p$ Monarch decomposition. Let $\omega(i)$ be a helper function that returns $\sigma_H$ if the intermediate result of step $i$ is stored on HBM, or $\sigma_S$ otherwise. The cost of the convolution using an order-$p$ Monarch decomposition is given by:

$$C = BH \sum_{i=1}^{p} \frac{16NN_i}{\gamma(N_i)} + \frac{4N}{\omega(i)} \tag{2}$$

Figure 3 graphs Equation 2 for different order-$p$ decompositions on different sequence lengths for A100, for $p \in \{2, 3, 4\}$. For cases where $N_1 = \cdots = N_p$, the total FLOP cost of an order-$p$ decomposition grows with $O(N^{(p+1)/p})$. However, for shorter sequences, higher-order decompositions are actually *more expensive*, since they decompose to matrices that are smaller than the matrix-matrix multiply unit (corresponding to the early bumps). Note also the bump in cost for $p = 3$ between 32K and 64K, which is a result of running out of SRAM but which is mediated by an extra decomposition for $p = 4$.

## 3.3 ARCHITECTURAL EXTENSIONS: SPARSITY IN CONVOLUTIONS

We present two architectural extensions to FLASHFFTCONV: partial convolutions and frequency-sparse convolutions, which are approximate convolution methods enabled by FLASHFFTCONV.

**Partial Convolutions** In partial convolutions, we zero out later portions of the convolution kernel, analogous to local attention. This has two benefits. First, it reduces the memory footprint, since it requires fewer elements to be held in GPU memory at once. Second, it allows for natural extensions of a pretrained convolutional model to longer sequences (i.e., via a sliding window approach).

**Frequency-Sparse Convolutions** In frequency-sparse convolutions, we zero out portions of the convolution kernel in frequency space, i.e. zeroing out portions of $k_f$. Here, the specific sparsity pattern can yield computational benefits—corresponding to skipping portions of the matrix multiplies in FLASHFFTCONV (see Appendix B for examples).

## 4 EXPERIMENTS

In this section, we evaluate FLASHFFTCONV in terms of quality and efficiency. First (Section 4.1), we show that FLASHFFTCONV allows models to achieve better quality for the same compute budget in language modeling—matching the performance of models with twice the parameters for free. FLASHFFTCONV also enables higher quality via higher resolution in image classification—solving the challenging Path-512 task for the first time simply via increased sequence length. Next (Section 4.2), we demonstrate FLASHFFTCONV's speedup for convolutions, evaluate its efficiency gains end-to-end, and compare a convolutional model using FLASHFFTCONV to Transformers using FlashAttention-v2. Finally (Section 4.3), we evaluate partial and frequency-sparse convolutions. Partial convolutions yield the first DNA model that can embed the longest genes at single nucleotide resolution (2.3M base pairs), and frequency-sparse convolutions yield speedup while maintaining—or improving—quality.

## 4.1 IMPACT OF EFFICIENCY ON QUALITY

We study how FLASHFFTCONV impacts downstream quality. First, given two implementations with the same compute budget, FLASHFFTCONV achieves higher quality due to higher training throughput. Second, we show that improved efficiency can lead to higher quality via longer sequence length.

**Improvement in Quality with Fixed Compute Budget** To evaluate the impacts of efficiency on downstream quality, we train two popular convolutional language models, M2-BERT-base (Fu et al., 2023a) and Hyena-s (Poli et al., 2023), from scratch. These models are trained BERT-style (masked language modeling) and GPT-style (next token prediction), respectively. We compare the quality of models trained with the same compute budget but different implementations of the convolution—either FLASHFFTCONV or a PyTorch implementation of the FFT convolution. FLASHFFTCONV achieves higher pretraining throughput, which allows the models to see more data during pretraining. These efficiency gains improve average GLUE score by up to 3.4 points for M2-BERT-base and perplexity

Table 1: Improvement in quality given a fixed compute budget.

| Model (Metric) | PyTorch | FLASHFFTCONV |
|---|---|---|
| M2-BERT-base-110M (GLUE Score ↑) | 77.6 | **80.9** |
| Hyena-s-155M (PPL ↓) | 13.4 | **11.1** |

Table 2: Classification accuracy (↑) on Path-X and Path-512 from the long range arena benchmark (Tay et al., 2020). FLASHFFTCONV allows for higher-resolution classification. ✗ indicates out of memory.

| Task (seq. len.) | PyTorch | FLASHFFTCONV |
|---|---|---|
| Path-X (16K) | **96.9** | **96.9** |
| Path-512 (256K) | ✗ | **96.1** |

Table 3: Time (↓) to compute the forward pass of a convolution with FLASHFFTCONV in milliseconds on one H100-SXM, as well as ablations removing specific optimizations. We also show memory savings. Results scaled to batch size 64, hidden dimension 768.

| | $p=2$ | | $p=3$ | | | | $p=4$ | | |
|---|---|---|---|---|---|---|---|---|---|
| Sequence Length | **256** | **1K** | **4K** | **8K** | **16K** | **32K** | **1M** | **2M** | **4M** |
| PyTorch | 0.43 | 1.57 | 6.65 | 13.7 | 28.6 | 62.1 | 2,346.3 | 4,892.1 | 10,127.6 |
| FLASHFFTCONV | **0.09** | **0.24** | **1.37** | **3.19** | **9.27** | **21.8** | **1,492.8** | **2,695.1** | **7,587.0** |
| Fusion-Only/cuFFTdx | 0.21 | 0.67 | 3.51 | 7.71 | 21.4 | 45.5 | – | – | – |
| Speedup over PyTorch | 4.78× | 6.54× | 4.85× | 4.29× | 3.09 × | 2.85× | 1.57× | 1.82× | 1.33× |
| Memory Savings | 8.21× | 7.73× | 7.61× | 7.59× | 7.21× | 6.57× | 2.64× | 2.63× | 2.63× |

by 2.3 points for Hyena-s. These improvements in quality are similar in magnitude to the effect of doubling the number of parameters in the model (see Appendix C for reference results).

**Longer Sequence Models** Next, we show how increased efficiency can lead to higher quality via longer sequence lengths. We evaluate long convolution models on Path-X and Path-512, high-resolution imaging tasks from the long range arena (LRA) benchmark (Tay et al., 2020).[1] These tasks take an image ($128 \times 128$ for Path-X and $512 \times 512$ for Path-512), flatten it out, and require a sequence model to classify whether two dots in the image are connected by a path.

Existing PyTorch implementations of convolutional sequence models (or even prior optimized implementations (Fu et al., 2023b)) fail to achieve better-than-random (50%) accuracy on Path-512 due to out of memory errors and a lack of support for such long sequences. Table 2 shows that FLASHFFTCONV allows a convolutional sequence model to solve Path-512 for the first time simply by increasing the available sequence length and reducing the memory footprint of the model through fusion.

## 4.2 EFFICIENCY

We evaluate FLASHFFTCONV on how fast it computes convolutions compared to PyTorch and how much speedup it yields for convolutional sequence models end-to-end. We also evaluate memory savings compared to PyTorch and compare against Transformers using FlashAttention-2 (Dao, 2023).

**FLASHFFTCONV Provides Speedup and Memory Savings** We benchmark the speed of the convolution compared against an FFT convolution implemented in PyTorch. We also benchmark ablations evaluating kernel fusion without using tensor cores—which recovers the strong baseline of using Nvidia's cuFFTdx kernel fusion library (NVIDIA, 2023a)—and FLASHFFTCONV without its domain-specific optimizations.

Table 3 shows that FLASHFFTCONV outperforms PyTorch FFT convolution across all sequence lengths, by up to 6.54×. Appendix C shows further speedup, up to 7.93×, for domain-specific optimizations. Speedups are greatest for short sequences, where the PyTorch FFT convolution is dominated by I/O costs. Speedup is more modest for longer sequences, which incur additional I/O costs (between registers and SRAM for the $p=3$ and between SRAM and HBM for $p=4$). Without using

---

[1]We refer to Path-512 as a scaled-up version of Path-256.

Table 4: End-to-end throughput ($\uparrow$) of convolutional sequence models against PyTorch.

| Model (size, seqlen, unit) | PyTorch | FlashFFTConv | Speedup |
|---|---|---|---|
| M2-BERT-base (110M, 128, seqs/s) | 4,480 | **8,580** | 1.9$\times$ |
| Hyena-s-4K (155M, 4K, seqs/s) | 84.1 | **147** | 1.7$\times$ |
| Long convs, Path-X (102M, 16K, images/s) | 126 | **308** | 2.4$\times$ |
| SaShiMi (5.4M, 64K, audio clips/s) | 38.7 | **50.3** | 1.3$\times$ |
| HyenaDNA (1M, seqs/s) | 0.69 | **3.03** | 4.4$\times$ |

Table 5: End-to-end throughput ($\uparrow$) in thousands of tokens per second, FLOP utilization, and speedup of Hyena against GPT running FlashAttention-v2 (Dao, 2023) across sequence lengths for A100.

| Model | 2K | 8K | 16K |
|---|---|---|---|
| GPT-2.7B, FA-v2 (Dao, 2023) | 33.8 | 27.8 | 21.6 |
| Hyena-2.7B, FLASHFFTCONV | **35.2** | **35.2** | **32.3** |
| FA-v2 FLOP Utilization | 65.7 | 72.1 | 78.5 |
| FLASHFFTCONV FLOP Utilization | 62.3 | 61.9 | 56.5 |
| FLASHFFTCONV Speedup | 1.1$\times$ | 1.3$\times$ | 1.5$\times$ |

the Monarch decomposition for tensor cores (fusion-only), FLASHFFTCONV becomes bottlenecked by the speed of general arithmetic operations on GPUs and the size of SRAM. Table 3 also shows memory savings over PyTorch. FLASHFFTCONV reduces the memory footprint of convolutions via recomputation in the backward pass and kernel fusion.

**FLASHFFTCONV Speeds Up Convolutional Sequence Models** We benchmark end-to-end throughput of convolutional sequence models across various modalities and sequence lengths spanning four orders of magnitude. We benchmark M2-BERT-base (Fu et al., 2023a), a BERT-style language model that has sequence length 128; Hyena-s-4K (Poli et al., 2023), a GPT-style language model with sequence length 4K; a long-convolutional model (Fu et al., 2023c) trained on Path-X with sequence length 16K (Tay et al., 2020); SaShiMi (Goel et al., 2022), an audio generation model trained on 1-second audio clips sampled at 64 KHz; and HyenaDNA-1M (Nguyen et al., 2023), a DNA modeling model trained on 1M sequence length. Details of the architectures and architecture-specific optimizations (such as fusing multiplicative gating for M2 and Hyena models) are given in Appendix D.

Table 4 shows that FLASHFFTCONV speeds up these models end-to-end. Speedup varies vary by the size of the models and the relative amount of time spent computing the convolution compared to other parts of the models. FLASHFFTCONV only speeds up the SaShiMi model by 1.3$\times$, since the model interleaves convolutions with SSM-based filter generation, pooling layers, and MLPs, which reduces the relative amount of time spent computing the convolution itself. Speedup is greatest for HyenaDNA, where PyTorch is bottlenecked by small batch size. The PyTorch implementation only allows batch size 1 on an 80GB GPU, whereas FLASHFFTCONV allows batch size 4—yielding significant speedup.

**FLASHFFTCONV is Faster than FlashAttention-v2** We compare end-to-end efficiency of a 2.7B-parameter Hyena model using FLASHFFTCONV against a 2.7B-parameter GPT model using FlashAttention-v2 (Dao, 2023) at three sequence lengths. Table 5 shows throughput, end-to-end FLOP utilization, and speedup. FLASHFFTCONV achieves lower end-to-end FLOP utilization than FlashAttention-v2 but achieves higher throughput, since convolutions incur fewer overall FLOPs.

## 4.3 Partial and Frequency-Sparse Convolutions

We evaluate the impact of partial convolutions on downstream quality and memory footprint and on how well they can extend the sequence length of existing models. We evaluate the impact of frequency-sparse convolutions on downstream quality, and we show that frequency-sparse convolutions can yield up to 1.4$\times$ additional speedup in the convolution without impacting quality.

**Partial Convolutions Reduce Memory Footprint and Increase Sequence Length** Partial convolutions reduce the memory footprint of models, in both language modeling and DNA modeling. A large proportion of the convolution filters can be pruned without impacting downstream quality. Table 6

Table 6: Quality and memory footprint of partial convolutions during training across sequence lengths.

| Hyena-s-8K | 8K | 4K | 2K | 1K | 512 | 256 |
|---|---|---|---|---|---|---|
| PPL (↓) | 13.8 | 13.8 | 13.8 | 13.9 | 14.0 | 14.2 |
| Memory Footprint (↓) | 32.5G | 15.3G | 11.8G | 8.4G | 6.1G | 5.8G |

Table 7: PPL (↓) from using partial convolutions to extend the sequence length of HyenaDNA to longer sequences. At 4M sequence length, the models are able to embed the longest human genes.

| Base Filter Length | 1M | 2M | 4M |
|---|---|---|---|
| HyenaDNA-450K | 2.91 | 2.91 | 2.91 |
| HyenaDNA-1M | 2.91 | 2.91 | 2.90 |

Table 8: Applying frequency-sparsity to the filters of a pretrained HyenaDNA-1M model.

| Sparsity Fraction | 0% | 50% | 75% | 79% | 84% | 91% |
|---|---|---|---|---|---|---|
| PPL (↓) | 2.91 | 2.91 | 2.90 | 2.91 | 2.93 | 2.98 |
| Convolution Speedup (↑) | 1.0× | 1.2× | 1.3× | 1.4× | 1.5× | 1.8× |

shows that a Hyena-s-8K model can be pretrained with a much shorter convolution kernel—as short as 2K—without negatively impacting quality.

Partial convolutions yield another benefit: we can naturally extend the sequence length of existing pretrained models. We extend a pretrained HyenaDNA-1M model to 4M sequence length with promising PPL results (Table 7)—yielding the first model that can embed the longest human genes at single-nucleotide resolution (2.3M base pairs) (See Appendix C for a visualization of gene embeddings).

**Frequency-Sparse Convolutions Increase Throughput** Frequency-sparse convolutions can increase the speed of convolutions—and may also have positive effects on quality. Table 8 shows that we can set up to 79% of the entries of the kernel $k_f$ to zero without losing quality. Sparsification in frequency space may even improve the quality of pretrained models slightly; the PPL of a pretrained HyenaDNA-1M model improves by 0.01 points after its kernels are 75% sparsified in frequency space—potentially as a result of removing high-frequency noise. Sparsification also yields up to $1.4\times$ speedup in the convolution via skipping entire blocks of the matrix-matrix multiplies in the Monarch decomposition. Appendix D provides more details about the sparsity patterns used in Table 8.

## 5 RELATED WORK

We provide an abbreviated summary of related work. An extended related work can be found in Appendix A. **Long convolutional models** have emerged as a promising alternative to Transformers for sequence modeling (Gu et al., 2021; Romero et al., 2021b; Poli et al., 2023; Ma et al., 2022; Hasani et al., 2022; Smith et al., 2023), particularly for for **long-context applications** such as DNA modeling and speech synthesis (Nguyen et al., 2022; Goel et al., 2022). Our work is related a long history of **efficient FFT algorithms**, ranging from Cooley-Tukey (Cooley & Tukey, 1965) to more recent FFT algorithms (Ayinala et al., 2011; Chu & George, 1999; Bahn et al., 2009; Bailey, 1989). Our work builds and extends these algorithms to adapt them to convolutions. Finally, partial and frequency-sparse convolutions relate to a long history of **sparsity in deep learning**, from pruning (Han et al., 2015a;b; Sanh et al., 2020) to sparse structured matrices (De Sa et al., 2018; Sindhwani et al., 2015).

## 6 CONCLUSION

We present FLASHFFTCONV, a new system for optimizing FFT convolutions for long sequences. We show that FLASHFFTCONV improves quality under a fixed compute budget, enables longer-sequence models, and improves the efficiency of long convolutions. We also show that analogues of sparsity in convolution filters map naturally on to FLASHFFTCONV's compute model, and can reduce memory footprint and runtime. We hope that our work will help support further adoption of convolutional sequence models, and that our insights can help inform the design of future architectures.

ACKNOWLEDGMENTS

We gratefully acknowledge the support of NIH under No. U54EB020405 (Mobilize), NSF under Nos. CCF2247015 (Hardware-Aware), CCF1763315 (Beyond Sparsity), CCF1563078 (Volume to Velocity), and 1937301 (RTML); US DEVCOM ARL under Nos. W911NF-23-2-0184 (Long-context) and W911NF-21-2-0251 (Interactive Human-AI Teaming); ONR under Nos. N000142312633 (Deep Signal Processing); Stanford HAI under No. 247183; NXP, Xilinx, LETI-CEA, Intel, IBM, Microsoft, NEC, Toshiba, TSMC, ARM, Hitachi, BASF, Accenture, Ericsson, Qualcomm, Analog Devices, Google Cloud, Salesforce, Total, the HAI-GCP Cloud Credits for Research program, the Stanford Data Science Initiative (SDSI), and members of the Stanford DAWN project: Meta, Google, and VMWare. The U.S. Government is authorized to reproduce and distribute reprints for Governmental purposes notwithstanding any copyright notation thereon. Any opinions, findings, and conclusions or recommendations expressed in this material are those of the authors and do not necessarily reflect the views, policies, or endorsements, either expressed or implied, of NIH, ONR, or the U.S. Government.

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

APPENDIX

We present extended related work (Appendix A), additional algorithmic details (Appendix B), additional experimental results (Appendix C), and experimental details (Appendix D).

## A  EXTENDED RELATED WORK

**Long Convolutions in Sequence Modeling**  Long convolutional models have emerged as a promising alternative to Transformers for sequence modeling (Gu et al., 2021; 2022a;b; Romero et al., 2021b;a; Poli et al., 2023; Ma et al., 2022; Fu et al., 2023c;b;a; Nguyen et al., 2023; Hasani et al., 2022; Smith et al., 2023). These methods differ in how they generate the convolutional kernels; for example, the S4 line of work uses learned state space models (Gu et al., 2021; Mehta et al., 2022; Gupta et al., 2022; Ma et al., 2022), while other works (Poli et al., 2023; Romero et al., 2021a;b) parameterize the convolution using an MLP from positional encodings. However, all the models operate by taking a convolution over the input sequence with a kernel as long as the input: $y = u * k$, where $u \in \mathbb{R}^{B \times H \times N}, k \in \mathbb{R}^{H \times N}$, and the kernel $k$ is broadcast along the $B$ dimension. When used for language modeling, these models often incorporate elementwise multiplicative gating as well: $y = f(u) \odot ((g(u) \odot h(u)) * k)$, where $f$, $g$, and $h$ are linear maps along the $H$ dimension (Poli et al., 2023; Fu et al., 2023b;a; Mehta et al., 2022; Wang et al., 2022a).

**Long-Context Applications**  Long convolutional models have especially been helpful for long-context applications, such as DNA modeling and speech synthesis. In DNA modeling, most longer-context genomic models have relied on either tokenization (Ji et al., 2021; Zaheer et al., 2020; Tay et al., 2021b) or downsampling (Fournier et al., 2021; Avsec et al., 2021). However, recent work has suggested that modeling DNA directly from base pairs can yield downstream improvements in quality, which requires long sequence lengths (Nguyen et al., 2023).

Like DNA modeling, speech synthesis has also benefited from long-context modeling. While traditional speech synthesis pipelines use intermediate representations such as spectrograms (Kumar et al., 2019; Prenger et al., 2019; Shen et al., 2018), linguistic features (Bińkowski et al., 2019; Kalchbrenner et al., 2018; Oord et al., 2016), or discrete audio codes (Dhariwal et al., 2020; Dieleman et al., 2018; Lakhotia et al., 2021; Van Den Oord et al., 2017), recent work has shown that modeling the speech directly from the raw waveform can yield downstream improvements in quality (Goel et al., 2022). Again, such models require long sequences to model audio at the rate at which it is naturally sampled, necessitating long-sequence modeling.

**FFT Algorithms**  There is a long history of efficient FFT algorithms, ranging from the Cooley-Tukey FFT algorithm published in 1965 (Cooley & Tukey, 1965) to parallel FFT algorithms (Ayinala et al., 2011) and more (Chu & George, 1999; Bahn et al., 2009; Bailey, 1989). These algorithms have enabled fundamental progress in a range of disciplines, from control theory (Brigham, 1988; Bekele, 2016) to signal processing (Oppenheim, 1978; Oppenheim et al., 2001). As FFTs prove more useful for modern deep learning applications, such as long convolutions, new techniques are required to run them efficiently on modern accelerators. Our work continues a line of work exploring how to use tensor cores for the FFT convolution (Fu et al., 2023c;b; Li et al., 2021), and extends the algorithmic capabilities to much longer sequences.

**Sparsity in Deep Learning**  As deep learning models have grown larger and deeper (Bommasani et al., 2021; Brown et al., 2020; Chowdhery et al., 2022), there is increasing interest in reducing the cost of training and running models. Sparsity in particular has received a great deal of attention, and has a long history in machine learning, including work in pruning neural networks (Han et al., 2015a;b; Sanh et al., 2020; Lin et al., 2017; Dong et al., 2017) and finding lottery tickets (Frankle & Carbin, 2018; Frankle et al., 2019; 2020). Our work in partial convolutions and frequency-sparse convolutions relates to this line of work, as an analogue of sparsity in convolutional filters. The Monarch decomposition is also closely related to structured matrices. Structured matrices have subquadratic ($o(n^2)$ for dimension $n \times n$) parameters and runtime, such as sparse and low-rank matrices, and fast transforms (Fourier, Chebyshev, sine/cosine, orthogonal polynomials) (Dao et al., 2022a). Structured matrices can often be computed with simple divide-and-conquer schemes, and can be used to represent many fast transforms (De Sa et al., 2018; Sindhwani et al., 2015; Kailath et al., 1979; Eidelman & Gohberg, 1999).

**Optimization of deep learning primitives**   There is a rich history of optimizing deep learning primitives. Many techniques, such as kernel fusion, aim to reduce data movement. Recently, libraries such as PyTorch 2.0 Paszke et al. (2019) have added kernel fusion automatically. Other techniques include checkpointing, wherein one stores fewer intermediate results and recomputes the others on-the-fly where they are needed, trading additional compute for memory Kusumoto et al. (2019); Wang et al. (2022b). Many algorithms also have hand-optimizations that can remove unnecessary computation or memory accesses Milakov & Gimelshein (2018).

Another line of optimization techniques aims to reduce FLOPs. MLPs and attention are particularly popular targets of FLOP reduction, via sparse factorizations of weights (Cooley & Tukey, 1965; Frankle & Carbin, 2018; Dao et al., 2022a; 2020; Zhu et al., 2021; Dao et al., 2021; Dettmers & Zettlemoyer, 2019; Chen et al., 2021a), or sparse/low-rank approximations of attention Beltagy et al. (2020); Kitaev et al. (2020); Ma et al. (2021); Fedus et al. (2022); Du et al. (2022); Wang et al. (2020); Dai et al. (2020); Zhu et al. (2021); Choromanski et al. (2020); Katharopoulos et al. (2020) and their combinations (Chen et al., 2021b; Tay et al., 2022).

# B    ALGORITHM DETAILS

## B.1    DOMAIN-SPECIFIC OPTIMIZATIONS

We review the details of how to compute a real-to-real FFT of size $N$ using a complex FFT of size $N/2$, following a tutorial by (Sorensen et al., 1987).

For this section, we adopt notation common in describing FFT algorithms. Let $x(n)$ be an input sequence of length $N$, and let $X(k)$ be the result of its discrete Fourier transform. Recall that:

$$X(k) = \sum_{n=0}^{N-1} x(n) W_N^{nk}, \tag{3}$$

for $k = 0, 1, ..., N-1$, where $W_N = e^{-2\pi i/N}$ is the $N$th root of unity.

First, if $x(n)$ is real, then symmetries emerge in $X(k)$. In particular, we have $X(k) = X^*(-k) = X^*(N-k)$, where $^*$ denotes complex conjugation. These symmetries allow us to have an algorithm for computing $X(k)$ using a single complex DFT of size $N/2$.

In particular:

$$X(k) = \sum_{n=0}^{N-1} x(n) W_N^{nk}$$
$$= \sum_{n=0}^{N/2-1} x(2n) W_{N/2}^{nk} + W_N^k \sum_{n=0}^{N/2-1} x(2n+1) W_{N/2}^{nk},$$

for $k = 0, 1, ..., N-1$. The DFT is now decomposed into two parts: a DFT over the even-indexed elements of $x(n)$, and over the odd-indexed elements of $x(n)$.

We can now create a third complex sequence, of length $N/2$, and put the even-indexed elements of $x(n)$ in the real part, and the odd-indexed elements of $x(n)$ in the imaginary part. Let:
$$z(n) = x(2n) + ix(2n+1),$$
for $n = 0, 1, ..., N/2-1$. Then, we compute the $N/2$-sized DFT $Z(k)$, and we can recover the DFT over the even and odd parts of $x(n)$ ($X_e[k]$ and $X_o[k]$, respectively):

$$X_e[k] = \frac{Z[k] + Z^*[N/2-k]}{2}$$
$$X_o[k] = -i\frac{Z[k] - Z^*[N/2-k]}{2i}.$$

We can now recover $X[k], k = 0...N-1$ using:
$$X[k] = X_e[k \mod N/2] + X_o[k \mod N/2] W_N^k.$$

The inverse FFT proceeds similarly. The goal is to recover $x(n)$ given an input $X[k]$, using a simple complex inverse DFT of length $N/2$.

First, we recover $X_e[k]$ and $X_o[k]$:

$$X_e[k] = \frac{X[k] + X^*[N/2 - k]}{2}$$
$$X_o[k] = \frac{X[k] - X^*[N/2 - k]}{2} W_N^k,$$

for $k = 0, ..., N/2 - 1$. Then, we construct $Z[k]$:
$$Z[k] = X_e[k] + iX_o[k], k = 0...,N/2 - 1.$$
We use the inverse DFT to recover $z(n)$, and then recover $x(n)$ from the real and imaginary parts of $z(n)$:

$$x(2n) = \text{Re}(z_n)$$
$$x(2n+1) = \text{Im}(z_n),$$

for $n = 0, ..., N/2 - 1$.

To implement these in our kernels, we perform the bookkeeping after reading the inputs or before writing the output, and then use the FFT/iFFT implementations as detailed in Algorithm 1 and others.

### B.2 LOW-LEVEL CUDA DETAILS

To ensure high performance, we implement CUDA kernels for each specific sequence length, allowing us to cater to specific performance nuances that arise from the decomposition at that sequence length. In this section, we dive into some of the low-level implementation details for FLASHFFTCONV.

**Matrix Multiplication Using CUDA Tensor cores**   CUDA Tensor cores can perform the multiplication of two $m \times k$ and $k \times n$ matrices for bfloat16 or float16 elements, using around the same number of cycles as is required for the multiplication of two scalars. $m \times k \times n$ must be of one of the following: $16 \times 16 \times 16$, $32 \times 8 \times 16$, $8 \times 32 \times 16$. This informs our choice of radix for decomposition when performing the FFT and iFFT. In particular our implementation breaks down matrix-matrix multiplications into blocked matrix-matrix multiplications where $m \times k \times n = 16 \times 16 \times 16$. We note the following about matrix-matrix multiplication on tensor cores (NVIDIA, 2023c):

- Tensor cores are utilized at the level of the warp and programmatic access of the tensor cores is via the Warp Level Matrix Multiply Accumulate (WMMA) API.

- Tensor core operands are held in register fragments ($wmma :: matrix\_a$, and $wmma :: matrix\_b$) and results are written to a register fragment ($wmma :: accumulator$).

- The operand fragments can hold data in row-major or column-major format and data in the $wmma :: accumulator$ fragment can be written to memory in row-major or column-major format.

- The specific mapping of items in a fragment to threads in warp is unspecified, however, the mapping of items to threads in the $wmma :: accumulator$ fragment exactly matches that for the $wmma :: matrix\_a$ fragment read row-major, allowing us to directly copy the results of a matrix-matrix multiplication and use as the operand for another matrix-matrix multiply.

To perform a matrix-matrix multiplication $C = A \times B$ using the tensor cores, a warp loads the contents of $A$ and $B$ into registers (WMMA fragments in CUDA parlance), performs the matrix-matrix multiplication, and writes the results which are stored in an accumulator fragment back to memory.

**Register Reuse**   A key part of ensuring high performance is minimizing I/O across different levels of the memory hierarchy: from HBM to SRAM and from SRAM to registers. To ensure this, we move the output from the $accumulator$ fragment directly into $matrix\_a$ fragment for use in subsequent matrix multiplications, avoiding an extra trip to SRAM. However, this is only possible if the output from the previous matrix-matrix multiply does not need to be transposed before using it as an operand for the next one. When this is not the case, we need to make a trip to SRAM and back. In Algorithm 2 we detail I/O from SRAM to registers.

**Locality and Tiling**   The algorithm is trivially parallelizable across $B$ and $H$, allowing us to tile in both dimensions at the threadblock level. In Algorithm 3 , all loops from $i \leftarrow 1$ to $N_1$ are warp-tiled.

---

**Algorithm 2** Detailed Annotation of FLASHFFTCONV core algorithm showing I/O from SRAM to register fragments, with two-way Monarch decomposition. We assume $N = N_1^2$ for simplicity here.

---

**Input:** Input $u \in \mathbb{R}^{B \times H \times N}$, convolution kernel $k_f \in \mathbb{C}^{H \times N}$, FFT matrices $\mathbf{F} \in \mathbb{C}^{N_1 \times N_1}$, $\mathbf{F^{-1}} \in \mathbb{C}^{N_1 \times N_1}$, Twiddle factors $t \in \mathbb{C}^N$, $t_{inv} \in \mathbb{C}^N$, $B$ tile size $B_{tile}$, $H$ tile size $H_{tile}$.
**Output:** Output $y \in \mathbb{R}^{B \times H \times N}$.
  **for** SMs in parallel across $B/B_{tile} \times H/H_{tile}$ **do**
      Load $\mathbf{F}, \mathbf{F^{-1}}, t, t_{inv}$ from HBM.
      **for** $h \leftarrow 1$ to $H_{tile}$ **do**
         Load $\mathbf{K_f} \leftarrow k_f[h]$ from HBM, reshaped to $N_1 \times N_1$.
         **for** $b \leftarrow 1$ to $B_{tile}$ **do**
            Load $\mathbf{X} \leftarrow u[b,h]$ from HBM, reshaped to $N_1 \times N_1$.
            $\mathbf{X} \leftarrow \mathbf{F}^\top \mathbf{X}$        $\triangleright \mathbf{F}^\top$ ($matrix\_a$), $\mathbf{X}$ ($matrix\_b$) output to $accumulator$
            Load $\mathbf{X}$ from $accumulator$ to $matrix\_a$
            $\mathbf{X} \leftarrow \mathbf{X} * t$         $\triangleright$ Elementwise multiply directly in $matrix\_a$
            $\mathbf{X} \leftarrow \mathbf{X}\mathbf{F}$        $\triangleright \mathbf{X}$ ($matrix\_a$), $\mathbf{F}$ ($matrix\_b$) output to $accumulator$
            Load $\mathbf{X}$ from $accumulator$ to $matrix\_a$
            $\mathbf{X} \leftarrow \mathbf{X} * \mathbf{K_f}^\top$      $\triangleright$ Elementwise multiply with $k_f$ directly in $matrix\_a$
            $\mathbf{X} \leftarrow \mathbf{X}\mathbf{F}^{-1}$      $\triangleright \mathbf{X}$ ($matrix\_a$), $\mathbf{F}^{-1}$ ($matrix\_b$) output to $accumulator$
            Write $\mathbf{X}$ from $accumulator$ fragment to SRAM
            Load $\mathbf{X}^\top$ from SRAM to $matrix\_a$ fragment
            $\mathbf{X} \leftarrow \mathbf{X}^\top * t_{inv}$      $\triangleright$ Elementwise multiply with $t_{inv}$ directly in $matrix\_a$
            $\mathbf{Y} \leftarrow \mathbf{X}\mathbf{F}^{-1}$      $\triangleright \mathbf{X}$ ($matrix\_a$), $\mathbf{F}^{-1}$ ($matrix\_b$) output to $accumulator$
            Write $\mathbf{Y}^\top$ to HBM.

---

**Miscellaneous optimizations**    In addition to the above optimizations, we also perform some other optimizations that provide marginal speedup. These include: utilizing vector intrinsics/types for performing memory reads/writes and arithmetic for 16-bit floating point (fp16) and brain float point (bf16), allowing non-tensor core operations on these types to be performed at around twice the normal speed. Furthermore, we double buffer I/O movements across all levels of the memory hierarchy, reducing warp stalls. We also aggressively tune our kernel hyperparameters such as block and tile dimensions, and loop unrolling factors for the best performance on the specific underlying hardware.

### B.3 GENERALIZATION TO 3-WAY AND 4-WAY MONARCH DECOMPOSITIONS

We provide algorithm listings for 3-way and 4-way Monarch Decompositions.

**3-Way Decomposition**    Algorithm 3 shows the algorithm for a 3-way Monarch decomposition. It involves one extra matrix multiply operation on either side of the FFT and iFFT, and proceeds over the algorithm in Algorithm 1 in an inner loop.

**4-way Decomposition**    For the 4-way decomposition, we assume that we need to write intermediate outputs to HBM. Here, we treat the 3-way decomposition as a sub-routine, and assume it has a fused kernel (i.e., Algorithm 3). We compute one matrix multiply for the FFT and one for the iFFT, and then call the kernel for the 3-way decomposition over the rows of the output. The algorithm is listed in Algorithm 4.

### B.4 FREQUENCY-SPARSE PATTERNS

We describe frequency-sparse patterns and the matmul savings in more detail here. We use the full 4-way decomposition case, since the algorithms generalize to lower-order decompositions.

Let $N = N_1^4$, and consider a kernel $k_f \in \mathbf{C}^N$. Consider the matrix multiply and looping operations that occur when computing the FFT portions of FLASHFFTCONV $(u, k_f)$ (the iFFT portions are the same, in the opposite order):

1. In Algorithm 4, there is one FFT operation over the columns of $u$, reshaped to $N_1 \times N/N_1$, and a Twiddle correction..

2. Then, Algorithm 3 iterates over the rows of $u$ for $\alpha := N_1$ steps.

**Algorithm 3** FLASHFFTCONV algorithm for 3-way decomposition. We assume $N = N_1^3$ for simplicity here.

---

**Input:** Input $u \in \mathbb{R}^{B \times H \times N}$, convolution kernel $k_f \in \mathbb{C}^{H \times N}$, FFT matrices $\mathbf{F} \in \mathbb{C}^{N_1 \times N_1}$, $\mathbf{F^{-1}} \in \mathbb{C}^{N_1 \times N_1}$, Twiddle factors $t_1 \in \mathbb{C}^{N_1^2}$, $t_{1,inv} \in \mathbb{C}_{\mathrel{\not\mathrel{k}}}^{N_1^2}$, $t_2 \in \mathbb{C}^N$, $t_{2,inv} \in \mathbb{C}^N$, $B$ tile size $B_{tile}$, $H$ tile size $H_{tile}$.

**Output:** Output $y \in \mathbb{R}^{B \times H \times N}$.

 **for** SMs in parallel across $B/B_{tile} \times H/H_{tile}$ **do**
  Load $\mathbf{F}$, $\mathbf{F^{-1}}$, $t$, $t_{inv}$ from HBM.
  **for** $h \leftarrow 1$ to $H_{tile}$ **do**
   Load $\mathbf{K_f} \leftarrow k_f[h]$ from HBM, reshaped to $N_1^2 \times N_1$.
   $\mathbf{K_f} \leftarrow K_f^T$.          ▷ Transpose last two dimensions.
   Reshape $\mathbf{K_f}$ to $N_1 \times N_1^2$.
   **for** $b \leftarrow 1$ to $B_{tile}$ **do**
    Load $\mathbf{X} \leftarrow u[b,h]$ from HBM, reshaped to $N_1 \times N_1 \times N_1$.
    **for** $i \leftarrow 1$ to $N_1$ **do**
     $\mathbf{X'} \leftarrow \mathbf{F X}[:, i*N_1 : (i+1)*N_1]$
     $\mathbf{X}[:, \mathbf{i}*\mathbf{N_1} : (\mathbf{i+1})*\mathbf{N_1}] \leftarrow \mathbf{X'}$    ▷ Transpose, matmul, transpose.
    $\mathbf{X} \leftarrow \mathbf{X} * t_2$
    **for** $i \leftarrow 1$ to $N_1$ **do**         ▷ Loop over rows
     $\mathbf{X'} \leftarrow \mathbf{F X}[i]$
     Reshape $\mathbf{X'}$ to $N_1 \times N_1$
     $\mathbf{X'} \leftarrow ((\mathbf{F^\top X'}) * t)\mathbf{F}$     ▷ FFT, decomposed into two steps
     $\mathbf{X'} \leftarrow \mathbf{X'} * \mathbf{K_f}[i]^\top$     ▷ Elementwise multiply with $k_f$
     $\mathbf{Y'} \leftarrow ((\mathbf{X' F^{-1}})^\top * t_{inv})\mathbf{F^{-1}}$  ▷ Inverse FFT, decomposed into two steps
     $\mathbf{Y'} \leftarrow \mathbf{Y'}^\top$
     $\mathbf{Y}[i] \leftarrow \mathbf{Y'}$          ▷ Finish inner loop
    $\mathbf{Y} \leftarrow \mathbf{Y} * t_{2,inv}$
    **for** $i \leftarrow 1$ to $N_1$ **do**
     $\mathbf{Y'} \leftarrow \mathbf{F Y}[:, i*N_1 : (i+1)*N_1]$
     $\mathbf{Y}[:, \mathbf{i}*\mathbf{N_1} : (\mathbf{i+1})*\mathbf{N_1}] \leftarrow \mathbf{Y'}$   ▷ Transpose, matmul, transpose.
    Write $\mathbf{Y}$ to HBM.

---

3. Let $u'$ be the row in a specific iteration. In Algorithm 3, there is an FFT over the columns of $u'$, reshaped to $N_1 \times N_1^2$, and a Twiddle correction.

4. Then, the inner loop iterates over the rows of $u'$ for $\beta := N_1$ steps.

5. In each loop, $u'$ has one FFT operation with a twiddle factor correction. Let the matrix of this FFT operation be denoted $\mathbf{A}$.

6. Then there is a second FFT operation. Let the matrix of this FFT operation be denoted $\mathbf{B}$.

Now, reshape $k_f$ to $N_1 \times N_1 \times N_1 \times N_1$. Let us consider how sparsity along the each of the four dimensions of $k_f$ lets us skip operations in the above steps.

- Sparsity in the first dimension allows us to skip computation in $\mathbf{B}$, exactly in proportion to how much of the first dimension we eliminate. This can result in cost savings, as long as $\mathbf{B}$ can still be expressed using the tensor cores on-chip after skipping the computation. For example, if $\mathbf{B}$ is $32 \times 32$, then $N_1 = 32$, and it does not make sense to eliminate more than half of the first dimension.

- Sparsity in the second dimension works exactly the same way, except it allows us to skip computation in $\mathbf{A}$.

- Sparsity in the third dimension lets us reduce $\beta$. Each row of the third dimension that we remove lets us skip one iteration of the inner loop in step 4 above.

- Sparsity in the fourth dimension lets us reduce $\alpha$. Each row of the fourth dimension that we remove lets us skip one iteration of the outer loop in step 2 above.

**Algorithm 4** FLASHFFTCONV algorithm for 4-way decomposition. We assume $N = N_1^4$ for simplicity here.

---

**Input:** Input $u \in \mathbb{R}^{B \times H \times N}$, convolution kernel $k_f \in \mathbb{C}^{H \times N}$, FFT matrices $\mathbf{F} \in \mathbb{C}^{N_1 \times N_1}$, $\mathbf{F^{-1}} \in \mathbb{C}^{N_1 \times N_1}$, Twiddle factors $t \in \mathbb{C}^N$, $t_{inv} \in \mathbb{C}_{\!\!\!/\!\!\!F}{}^N$, $t_2 \in \mathbb{C}^N$, $t_{2,inv} \in \mathbb{C}^N$.

**Output:** Output $y \in \mathbb{R}^{B \times H \times N}$.

   Reshape $u$ to $B \times H \times N_1 \times (N/N_1)$.
   Reshape $k_f$ to $H \times N_1 \times (N/N_1)$.
   $k_f \leftarrow k_f^\top$.                                             ▷ Transpose last two dimensions.
   Reshape $k_f$ to $HN_1 \times N/N_1$.
   $u \leftarrow \mathbf{F}u$                                ▷ Computes the FFT over the columns of $u$.
   Reshape $u$ to $B \times (HN_1) \times (N/N_1)$.              ▷ Move $N_1$ into $H$ dimension.
   Reshape $k_f$ to $(HN_1) \times (N/N_1)$.
   Call FLASHFFTCONV $(u, k_f)$.                  ▷ Call 3-way FLASHFFTCONV.
   Reshape $u$ to $B \times H \times N_1 \times (N/N_1)$.
   $y \leftarrow \mathbf{F^{-1}}u$                            ▷ Computes the iFFT over the columns of $u$.
   Return $y$.

---

Table 9: Sparsity patterns for $k_f$ and sparsity fraction for the frequency-sparse convolution experiment in Table 8.

| Sparsity Pattern | S |
|---|---|
| a=0,b=0,c=0,d=0 | 0 |
| a=16,b=0,c=0,d=0 | 50 |
| a=16,b=16,c=0,d=0 | 75 |
| a=16,b=16,c=4,d=4 | 79 |
| a=16,b=16,c=8,d=8 | 84 |
| a=16,b=16,c=16,d=16 | 91 |

Table 10: Time ($\downarrow$) to compute the forward pass of a gated convolution with FLASHFFTCONV in milliseconds on one H100-SXM. We also show memory savings. All results scaled to batch size 64, hidden dimension 768. $p$ indicates the order of the Monarch decomposition.

| | $p=2$ | | $p=3$ | | | | $p=4$ | | |
|---|---|---|---|---|---|---|---|---|---|
| Sequence Length | **256** | **1K** | **4K** | **8K** | **16K** | **32K** | **1M** | **2M** | **4M** |
| PyTorch | 0.62 | 2.30 | 9.49 | 19.4 | 29.9 | 84.8 | 3,071.4 | 6,342.6 | 13,031.2 |
| FLASHFFTCONV | **0.11** | **0.29** | **1.43** | **3.58** | **12.2** | **26.3** | **1,768.9** | **4,623.5** | **10,049.4** |
| Speedup | 5.64× | 7.93× | 6.64× | 5.42× | 2.45× | 3.22× | 1.74× | 1.37× | 1.30× |
| Memory Savings | 6.65× | 6.40× | 6.35× | 6.34× | 6.17× | 5.87× | 2.82× | 2.81× | 2.81× |

As an example, we reveal the sparsity dimensions that we applied in the experiment detailed in Table 8 in the main paper. Conceptually, we use the full 2-million length kernel $k_f$, and reshape it to $32 \times 32 \times 32 \times 64$. Let $a$, $b$, $c$, and $d$ be variables describing how much of each dimension we set to zero. Specifically, we set $k_f[a:,:,:,:] = 0$, $k_f[:,b:,:,:] = 0$, $k_f[:,:,c:,:] = 0$, and $k_f[:,:,:,d:] = 0$ sequentially. The formula the sparsity fraction $S$ given $a,b,c,d$ in this case is given by:

$$S = 1 - (32-a)(32-b)(32-c)(64-d),$$

or more generally, 1 minus the product of the fraction of each dimension that is removed. Table 9 lists the configurations of the sparsity patterns and the sparsity fractions used for the experiment in Table 8.

### B.5 HARDWARE SUPPORT

FLASHFFTCONV was developed on A100 GPUs, and tested on A100 and H100 GPUs. Older generations of GPU such as V100 are not supported, since the sizes of the tensor cores are different. We look forward to integrating more general libraries such as Cutlass (NVIDIA, 2023b) to support a wider range of GPUs, and developing support for non-GPU accelerators.

## C ADDITIONAL RESULTS

Table 11: Full results for the forward pass of a convolution with FLASHFFTCONV compared to PyTorch in milliseconds on one H100-SXM. Batch size 64, hidden dimension 768.

| Seq Len | PyTorch | FLASHFFTCONV | Speedup |
|---|---|---|---|
| 256 | 0.43 | 0.09 | 4.69 |
| 512 | 0.81 | 0.15 | 5.34 |
| 1024 | 1.57 | 0.24 | 6.61 |
| 2048 | 3.27 | 0.55 | 5.95 |
| 4096 | 6.65 | 1.37 | 4.87 |
| 8192 | 13.72 | 3.19 | 4.30 |
| 16384 | 28.58 | 9.27 | 3.09 |
| 32768 | 62.09 | 21.84 | 2.84 |
| 65536 | 141.15 | 67.96 | 2.08 |
| 131072 | 292.26 | 147.26 | 1.98 |
| 262144 | 582.76 | 308.48 | 1.89 |
| 524288 | 1,167.28 | 742.26 | 1.57 |
| 1048576 | 2,346.26 | 1,492.84 | 1.57 |
| 2097152 | 4,892.09 | 2,695.51 | 1.81 |
| 4194304 | 10,127.56 | 7,586.96 | 1.33 |

Table 12: Full results for the forward pass of a gated convolution with FLASHFFTCONV compared to PyTorch in milliseconds on one H100-SXM. Batch size 64, hidden dimension 768.

| Seq Len | PyTorch | FLASHFFTCONV | Speedup |
|---|---|---|---|
| 256 | 0.62 | 0.11 | 5.76 |
| 512 | 1.18 | 0.19 | 6.14 |
| 1024 | 2.30 | 0.29 | 7.81 |
| 2048 | 4.70 | 0.67 | 7.05 |
| 4096 | 9.49 | 1.43 | 6.65 |
| 8192 | 19.38 | 3.58 | 5.42 |
| 16384 | 39.91 | 12.18 | 3.28 |
| 32768 | 84.79 | 26.32 | 3.22 |
| 65536 | 186.69 | 79.84 | 2.34 |
| 131072 | 382.98 | 181.51 | 2.11 |
| 262144 | 764.08 | 376.96 | 2.03 |
| 524288 | 1,530.34 | 878.93 | 1.74 |
| 1048576 | 3,071.37 | 1,768.94 | 1.74 |
| 2097152 | 6,342.58 | 4,623.46 | 1.37 |
| 4194304 | 13,031.21 | 10,049.42 | 1.30 |

## C.1 SPEEDUP FROM DOMAIN-SPECIFIC OPTIMIZATIONS

We benchmark domain-specific optimizations in FLASHFFTCONV. Table 10 shows the performance of a gated convolution $y = v \odot ((u \odot w) * k)$, where $v$ and $w$ are linear projections of the input $u$. This pattern is common in convolutional and SSM-based architectures for language modeling (Fu et al., 2023c;b; Poli et al., 2023; Mehta et al., 2022). A PyTorch implementation of a gated convolution incurs additional I/O overhead from the gating operations, whereas FLASHFFTCONV fuses the gating operations into the convolution. This fusion results in further speedup over PyTorch, up to $7.93\times$.

## C.2 FULL RESULTS FOR ALL SEQUENCE LENGTHS AND SETTINGS

We report full results for all sequence lengths in powers of two between 256 and 4M. We report full results for five cases:

- Table 11: Standard forward pass, where the FFT size is the same as the input size. This is equivalent to a circular convolution.

- Table 12: Gated forward pass, where the FFT size is the same as the input size.

Table 13: Full results for the forward pass of a convolution where the input is half the length of the convolution size with FLASHFFTCONV compared to PyTorch in milliseconds on one H100-SXM. Batch size 64, hidden dimension 768.

| Seq Len | PyTorch | FLASHFFTCONV | Speedup |
|---|---|---|---|
| 256 | 0.44 | 0.09 | 4.64 |
| 512 | 0.82 | 0.16 | 5.03 |
| 1024 | 1.57 | 0.24 | 6.45 |
| 2048 | 3.25 | 0.53 | 6.08 |
| 4096 | 6.59 | 1.37 | 4.83 |
| 8192 | 13.60 | 3.13 | 4.34 |
| 16384 | 28.37 | 8.82 | 3.22 |
| 32768 | 61.87 | 21.34 | 2.90 |
| 65536 | 141.42 | 77.32 | 1.83 |
| 131072 | 292.26 | 151.28 | 1.93 |
| 262144 | 582.82 | 315.99 | 1.84 |
| 524288 | 1,167.21 | 757.33 | 1.54 |
| 1048576 | 2,343.55 | 1,525.13 | 1.54 |
| 2097152 | 4,922.63 | 3,321.71 | 1.48 |
| 4194304 | 10,179.86 | 7,305.61 | 1.39 |

Table 14: Full results for the forward pass of a gated convolution where the input is half the length of the convolution size with FLASHFFTCONV compared to PyTorch in milliseconds on one H100-SXM. Batch size 64, hidden dimension 768.

| Seq Len | PyTorch | FLASHFFTCONV | Speedup |
|---|---|---|---|
| 256 | 0.54 | 0.11 | 4.71 |
| 512 | 1.01 | 0.19 | 5.27 |
| 1024 | 1.94 | 0.29 | 6.75 |
| 2048 | 3.97 | 0.59 | 6.69 |
| 4096 | 8.01 | 1.41 | 5.68 |
| 8192 | 16.42 | 3.46 | 4.75 |
| 16384 | 34.04 | 10.62 | 3.21 |
| 32768 | 73.15 | 25.03 | 2.92 |
| 65536 | 163.75 | 78.88 | 2.08 |
| 131072 | 337.37 | 153.13 | 2.20 |
| 262144 | 672.48 | 319.47 | 2.10 |
| 524288 | 1,346.99 | 763.97 | 1.76 |
| 1048576 | 2,704.91 | 1,538.89 | 1.76 |
| 2097152 | 5,644.20 | 3,545.79 | 1.59 |
| 4194304 | 11,625.79 | 8,132.32 | 1.43 |

- Table 13: Forward pass, where the input size is half the FFT size. This is equivalent to a causal convolution.

- Table 14: Gated forward pass, where the input size is half the FFT size.

- Table 15 Standard backward pass, where the FFT size is the same as the input size.

- Table 16 Memory use for FLASHFFTCONV compared to PyTorch for a convolution, scaled to batch size 64, hidden dimension 768.

- Table 17 Memory use for a gated convolution using FLASHFFTCONV compared to PyTorch for a convolution, scaled to batch size 64, hidden dimension 768.

Speedups vary, but generally follow the trend from the results in the body of the paper. FLASHFFT-CONV achieves significant memory savings over PyTorch due to recomputation in the backward pass and kernel fusion. To measure memory savings, we measure the relative additional memory from calling the convolution operations (we do not measure the footprint of hte original inputs).

Table 15: Full results for the backward pass of a convolution with FLASHFFTCONV compared to PyTorch in milliseconds on one H100-SXM. Batch size 64, hidden dimension 768.

| Seq Len | PyTorch | FLASHFFTCONV | Speedup |
|---------|---------|--------------|---------|
| 256 | 0.76 | 0.24 | 3.24 |
| 512 | 1.45 | 0.22 | 6.43 |
| 1024 | 2.83 | 0.65 | 4.37 |
| 2048 | 5.76 | 1.48 | 3.90 |
| 4096 | 11.56 | 2.86 | 4.05 |
| 8192 | 23.11 | 6.16 | 3.75 |
| 16384 | 46.85 | 18.57 | 2.52 |
| 32768 | 103.85 | 57.68 | 1.80 |
| 65536 | 241.81 | 111.76 | 2.16 |
| 131072 | 489.38 | 239.32 | 2.04 |
| 262144 | 976.24 | 519.49 | 1.88 |
| 524288 | 1,960.31 | 1,240.95 | 1.58 |
| 1048576 | 3,938.92 | 2,708.36 | 1.45 |
| 2097152 | 7,909.27 | 4,977.93 | 1.59 |
| 4194304 | 16,552.21 | 12,932.02 | 1.28 |

Table 16: Memory usage in GB for FLASHFFTCONV compared to PyTorch. Scaled up to batch size 64, hidden dimension 768.

| Seq Len | PyTorch | FLASHFFTCONV | Memory Reduction |
|---------|---------|--------------|------------------|
| 256 | 0.42 | 0.05 | 8.21× |
| 512 | 0.80 | 0.10 | 8.19× |
| 1024 | 1.58 | 0.20 | 7.73× |
| 2048 | 3.12 | 0.39 | 7.94× |
| 4096 | 6.21 | 0.82 | 7.61× |
| 8192 | 12.39 | 1.63 | 7.59× |
| 16384 | 24.93 | 3.46 | 7.21× |
| 32768 | 50.43 | 7.68 | 6.57× |
| 65536 | 121.60 | 46.08 | 2.64× |
| 131072 | 243.21 | 92.18 | 2.64× |
| 262144 | 486.41 | 184.39 | 2.64× |
| 524288 | 972.83 | 368.91 | 2.64× |
| 1048576 | 1945.65 | 738.34 | 2.64× |
| 2097152 | 3889.23 | 1477.69 | 2.63× |
| 4194304 | 7778.45 | 2961.56 | 2.63× |

## C.3 REFERENCE LARGER MODELS

Table 18 gives performance numbers for larger models trained for the same number of tokens and steps as the reference PyTorch models in Table 1 in the main paper.

The GPT-style PyTorch models are trained for 5B tokens, with batch size 512K tokens. The BERT-style PyTorch models are trained for 16000 steps, with batch size 64K tokens. In contrast, the FLASHFFTCONV models, with higher training throughput, are trained for 15B tokens and 70000 steps in the same compute budget, respectively.

## C.4 DNA EMBEDDINGS

We use our 4M-sequence length HyenaDNA model to generate embeddings for various DNA segments following the procedure from (Nguyen et al., 2023). The DNA classes include human genes corresponding to different biological function annotations from the Ensembl genome dataset known as biotypes (Cunningham et al., 2022). The longest human gene, the dystrophin gene, is annotated.

Table 17: Memory usage in GB for FLASHFFTCONV for a gated convolution compared to PyTorch. Scaled up to batch size 64, hidden dimension 768.

| Seq Len | PyTorch | FLASHFFTCONV | Memory Reduction |
|---|---|---|---|
| 256 | 0.66 | 0.10 | 6.65× |
| 512 | 1.28 | 0.19 | 6.61× |
| 1024 | 2.54 | 0.40 | 6.40× |
| 2048 | 5.04 | 0.78 | 6.49× |
| 4096 | 10.05 | 1.58 | 6.35× |
| 8192 | 20.07 | 3.17 | 6.34× |
| 16384 | 40.29 | 6.53 | 6.17× |
| 32768 | 81.15 | 13.83 | 5.87× |
| 65536 | 164.61 | 58.37 | 2.82× |
| 131072 | 329.22 | 116.75 | 2.82× |
| 262144 | 658.44 | 233.54 | 2.82× |
| 524288 | 1316.89 | 467.21 | 2.82× |
| 1048576 | 2633.78 | 934.95 | 2.82× |
| 2097152 | 5265.48 | 1870.90 | 2.81× |
| 4194304 | 10530.97 | 3747.99 | 2.81× |

Table 18: Reference quality numbers for models when trained for the same number of steps and training data.

| Model (Metric) | |
|---|---|
| M2-BERT-base-110M (GLUE Score ↑) | 77.6 |
| M2-BERT-large-260M (GLUE Score ↑) | 81.0 |
| Hyena-s-155M (PPL ↓) | 13.4 |
| Hyena-m-355M (PPL ↓) | 11.1 |

Table 19: $L_\infty$ error for FLASHFFTCONV compared to PyTorch FFT convolution in fp32 across sequence lengths, for fp16 and bf16.

| | 256 | 1K | 4K | 32K | 1M |
|---|---|---|---|---|---|
| fp16 | 3.8e-6 | 7.6e-6 | 1.5e-5 | 6.5e-5 | 4.0e-4 |
| bf16 | 3.1e-5 | 3.1e-5 | 6.1e-5 | 6.1e-5 | 6.1e-5 |

Table 20: Time spent in the convolution operation for the models in Table 4, on H100.

| Model | % Time in Convolution | Speedup |
|---|---|---|
| M2-BERT | 48.5% | 1.9x |
| Hyena | 38.4% | 1.7x |
| Long Convs | 64.3% | 2.4x |
| SaShiMi | 22.1% | 1.3x |
| HyenaDNA | 81.8% | 4.4x |

## C.5 PRECISION OF FLASHFFTCONV

FLASHFFTCONV operates over inputs in fp16 or bf16, which enables faster execution (tensor cores $2\times$ faster than fp32 matrix multiply) at the cost of lower-precision computation. We measure the error from computing an FFT convolution using FLASHFFTCONV compared to computing it in PyTorch in fp32 precision. Table 19 shows the results, using an $L_\infty$ norm. Error increases with sequence length, but is still small even for sequences up to 1M.

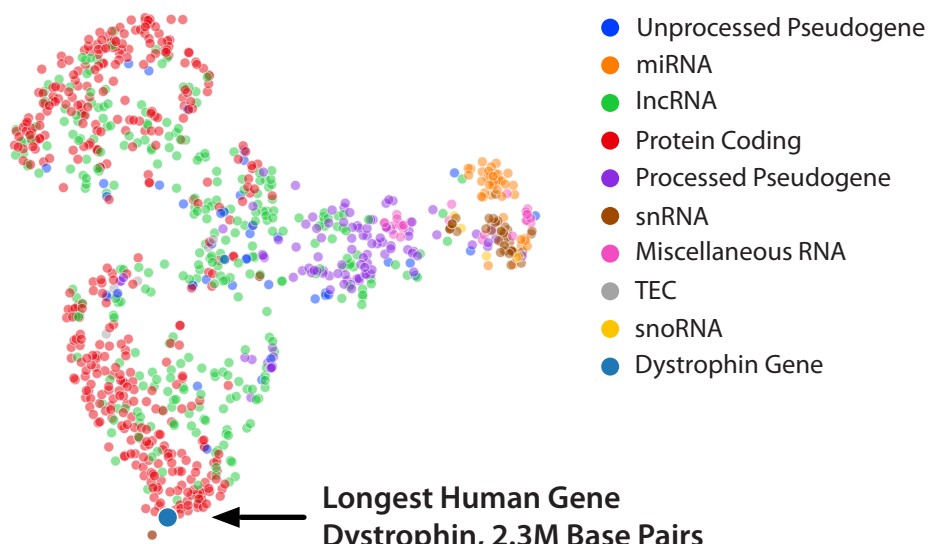

Figure 4: t-SNE visualization of various genes and DNA segments using our new HyenaDNA-4M. The longest human gene, Dystrophin, is annotated.

### C.6   PERCENTAGE OF TIME SPENT IN CONVOLUTION

Table 4 shows different amounts of speedup for different models end-to-end. Speedup varies by the amount of time each model spends computing the convolution, compared to other operations like MLP and convolution kernel generation.

Table 20 presents the amount of time a PyTorch implementation of each model spends computing convolutions. We profile each model on H100 using the PyTorch CUDA profiler and manually inspect the amount of time spent in the convolution. Speedup is closely correlated with how much time the PyTorch implementation spends in convolution.

## D   EXPERIMENT DETAILS

### D.1   COMPUTE

All experiments were conducted on a box with 8xA100-40GB GPUs or a box with 8xH100-SXM GPUs.

### D.2   FIXED COMPUTE BUDGET EXPERIMENT

For the experiment in Table 1, we train an M2-BERT-base model from scratch, and a Hyena-s-155M model from scratch.

We train the M2-BERT-base model using masked language modeling of 30% on the C4 dataset, and fine-tune it on GLUE using the protocol from (Fu et al., 2023a). The FLASHFFTCONV model has higher training throughput, so it trains for more tokens; we train the FLASHFFTCONV model for 70,000 steps with a batch size of 64K tokens. The PyTorch model, with lower training throughput, only trains for 16,000 steps, with the same batch size. The M2-BERT-base model we use is parameter-matched with a Transformer BERT-base. It has 12 hidden layers, with a model dimension of 960, and an expansion factor of four. It also uses a block-diagonal MLP with four blocks. The M2 Hyena filter has embedding dimension 5, filter order 128, and initial sine activation factor of 10. We train with learning rate 8e-4, weight decay 1e-5, and 6% warmup with a cosine decay.

We train the Hyena-s-155M model using a causal language modeling objective on the Pile. We train the FLASHFFTCONV model for 15M tokens, and the PyTorch model for 5M tokens. The Hyena-s-155M model matches the configuration from (Poli et al., 2023) and has 18 layers, with a hidden dimension of 864, and an expansion factor of 4. The Hyena filter has embedding dimension 33, filter order 64, and

initial sine activation factor of 14. We train with learning rate 6e-4, with 1% warmup time and a cosine decay.

### D.3 PATH-X AND PATH-512 EXPERIMENTS

For the experiment in Table 2, we use simple convolutional language models, as in (Fu et al., 2023c).

For Path-X, we use the same model and hyperparameters as the convolutional model from (Fu et al., 2023c). We use a convolutional model with 6 layers, prenorm batch norm, and hidden dimension of 256. For the convolution filter parameters, we use kernel dropout 0.3, kernel learning rate 0.0005, $\lambda$ factor 0.001, and two channels on the filter. We use an overall learning rate of 0.0005 and weight decay 0.05. We train for 500000 steps, with 10000 steps of warmup with a cosine decay, and global batch size 16.

For Path-512, we scale up the resolution of Path-256. We train for 200000 steps, with 10000 steps warmup, learning rate 0.0005, and weight decay 0.05. For the model, we train with 4 layers, and hidden dimension 256. We use kernel dropout 0.1, kernel learning rate 0.0005, $\lambda$ factor 0.001, and two channels on the filter. We keep the filter length to be 65536.

### D.4 CONVOLUTION BENCHMARKS

For the experiments in Table 3, we time the forward pass of a convolution with batch size 64, hidden dimension 768, and varying sequence length. If we run out of memory for a sequence length, we split the batch and hidden dimension and call the forward pass multiple times. We time each call 30 times and take the average of the runs. We use the same protocol for the backward pass in Table **??**.

### D.5 END-TO-END MODELING DETAILS

For the experiments in Table 4, we run forward pass of each model, and use it to compute throughput. Batch sizes vary by model, and we check throughput calculations with a few batch sizes to make sure the result is consistent. For the M2-BERT-base model, we use a 110M model from Monarch Mixer (Fu et al., 2023a). For the Hyena-s-4K model, we use an identical model to the one in Table 1, but with a filter length of 4K. For the long convs Path-X model, we use the same model as in Table 2. For the SaShiMi model, we use the standalone SaShiMi model from the official implementation (Goel et al., 2022), and we use 8 layers with hidden dimension 64, and 4 up pool and down pool layers. For the HyenaDNA model, we use the official 1M-sequence length checkpoint from (Nguyen et al., 2023). For M2-BERT-base, Hyena-s-4K, and HyenaDNA, we additionally use a fast depthwise convolution kernel for short kernels. For M2-BERT-base, Hyena-s-4K, and HyenaDNA, we report results benchmarked on one H100-SXM. For the others, we report performance on one A100-40GB.

### D.6 COMPARISON TO TRANSFORMERS

For the comparison against Transformers in Table 5, we use the official implementations with the FlashAttention-v2 release (Dao, 2023). We use a Hyena model, and match the number of layers, hidden dimension, and expansion factor to the 2.7B Transformer model. To compute the FLOP usage, we take the formula:

$$2*\text{num tokens}*\text{num parameters}$$

for the parametric FLOPs. For the non-parameter FLOPs, we add the raw FLOP count from our cost model in Equation 2 (without the adjustment for speed of tensor core FLOPs).

### D.7 PARTIAL CONVOLUTIONS FOR HYENA

For the measurement of memory footprint reduction in Table 6, we use the same Hyena-s model as in Tables 1 and 4, except we cut the filter short. This lets us offload parts of the input, which reduces the memory footprint.

### D.8 EXTENDING HYENADNA-1M

In Table 7, we use a sliding window approach to extend the HyenaDNA-1M and HyenaDNA-450K models to longer sequences. This mimics training a 4M-sequence HyenaDNA with a short filter.

### D.9 FREQUENCY-SPARSE CONVOLUTIONS

To evaluate frequency-sparse convolutions, we take the pretrained HyenaDNA-1M model, and sparsify $k_f$ using the strategy described in Appendix B.4. We then run standard validation using the validation set from (Nguyen et al., 2023).

Table 21: Measured Constants for Cost Model for A100-40GB.

| Constant | A100-40GB |
|----------|-----------|
| $\sigma_H$ | 1.35 TB/s |
| $\sigma_S$ | 9.5 TB/s |
| $\tau_M$ | 234 TFLOPs |
| $\tau_G$ | 17.6 TFLOPs |

### D.10 EMPIRICAL GPU PROFILING

Table 21 gives empirically-measured GPU stats for an A100-40GB, which we used to generate Figure 3. The statistics are specialized to the Monarch decomposition workload. To measure the achievable tensor core FLOPs, we measured the utilization of real fp16 matrix multiply. To measure achievable general arithmetic FLOPs, we measured the utilization of continuously applying Twiddle factors. To measure the achievable HBM bandwidth, we measured the speed of `torch.clone` of a tensor. To measure the achievable SRAM bandwidth, we measured the slow down from writing intermediate results to SRAM between matrix multiply instructions.

