# OpenReview forum: "FlashFFTConv: Efficient Convolutions for Long Sequences with Tensor Cores"
_ICLR.cc/2024/Conference — ICLR 2024 poster_

### Official Review · Reviewer_8BY8 · 2023-10-31

**Soundness:** 3 good
**Presentation:** 3 good
**Contribution:** 3 good
**Rating:** 6
**Confidence:** 5

**Summary:**

The paper proposes FlashFFTConv, a new algorithm for computing FFT on GPUs. This algorithm is more efficient than existing methods and can be used to accelerate a variety of machine learning tasks, especially for the long-sequence tasks. The author proposes approximation algorithms by leveraging the sparsity.

**Strengths:**

1. The paper is well organized and easy to follow.
2. The implementation is in the supplementary material. The appendices provide many details.
3. The method is solid and the experiments are convincing.

**Weaknesses:**

### Major issues
1. My understanding is that the main algorithm is arithmetic equivalent to the standard FFT. In other words, the proposed methods can be seen as an implementation of the FFT without any approximations. The method in Section 3.3 is an approximation algorithm. Is it correct?
2. FlashAttention is a great success since the attention is a "new" operator and lack efficient implementation. However, FFT is a standard operator with long history. Specifically, the Monarch FFT Decomposition was developed in the last century. Considering that this algorithm is not new and GPU engineers can handle the memory hierarchy and other hardware specifications properly, why this method was not developed previously?
I do not know the detailed algorithms in the cuFFT library, which is assumed to be highly-optimized. What is the reason of better performance of FlashFFTConv over cuFFT, (1) the better algorithm, i.e., Monarch FFT Decomposition, or (2) a better implementation considering the GPU architecture?
3. In Table 3, FlashFFTConv outperforms torch.fft by up to 8.7×, while the speedup is about 2x without the domain-specific optimizations. Does it mean the major speedup comes from the domain-specific optimizations instead of the FlashFFTConv algorithm? Could the authors conduct this ablation study (with and without the domain-specific optimizations) in other experiments?
4. When analyzing Table 4, the authors claim that "speedup varies by the size of the models and the relative amount of time spent computing the convolution compared to other parts of the models". Please provide the quantitative results, e.g., the relative amount of time spent computing the convolution.

### Minor issues
1. What about the results on small and medium sequence tasks?
2. Other than the machine learning applications, FFT is widely used in many fields. It is better to expand the scope of the paper. What about the results in signal processing?

**Questions:**

1. What are the limitations and extensions of the method?
2. What is the potential negative impact of the method? Can we always obtain better performance on FlashFFTConv over cuFFT?

---

> ### Author Response · Authors · 2023-11-16
> **Response to Reviewer 8BY8**
>
> We thank the reviewer for their thoughtful comments and constructive feedback. Below, we answer specific questions about which algorithms are exact and which are approximate, and where the speedup comes from on different applications.
>
> **Approximation Algorithms**
> We clarify that there are actually **three algorithmic contributions** in our paper. The main contribution is the FlashFFTConv algorithm, which is an **exact** FFT convolution algorithm.
>
> But the structure of the FlashFFTConv algorithm allows immediate applications to sparsity – simply skipping portions of the matrix multiply operations yields sparse convolution algorithms. Partial convolutions and frequency-sparse convolutions are two approximate convolution algorithms that are enabled by the computational pattern of FlashFFTConv.
>
> **Why is this not in cuFFT?**
> In short – a single FFT on its own is almost always memory-bound, so the most efficient FFT implementation does not need to use tensor cores or other optimization. In other words, the older FFT algorithms suffice. These are what cuFFT uses. Please see the common response for a more detailed discussion on the history of FFT algorithms, and why the FFT convolution presents specific bottlenecks.
>
> When fusing multiple FFT operations together, the entire operation becomes compute-bound, so different algorithms are necessary – which is why we need to translate the operation to use tensor cores.
>
> Finally, we note that the use case of long convolutions has started to show promise in ML applications only relatively recently. For example, S4 was only published at ICLR last year, and gated convolutional models that come close to attention performance in language modeling have only started to emerge this year. So compared to FlashAttention, which took 5 years between the introduction of attention and the first IO-aware algorithm, FlashFFTConv is moving relatively quickly!
>
> **More Detailed Breakdown of Speedup**
> Thank you for the feedback to conduct a more detailed ablation study of the speedups. We have re-organized our convolution benchmarks to several distinct use cases:
> * A pure convolution, where the input and FFT size are the same length
> * A convolution with padding, where the input is half the length of the FFT size (i.e., causal convolution)
> * Gated convolutions, where there are gating operations before and after the convolution to fuse
>
> We have chosen to split the benchmarks into these distinct use cases, instead of a single table with “domain-specific optimizations.” The full results are given in Appendices C.1 and C.2 of the updated draft.
>
> **Percentage of Time Each Model Spends Computing the Convolution**
> We provide the percentage of time each model spends computing the convolution when implemented in PyTorch, and the relative speedup from using FlashFFTConv:
>
> |      **Model** | **% Time in Convolution** | **Speedup** |
> |---------------:|:-------------------------:|:-----------:|
> |    **M2-BERT** |           48.5%           |     1.9x    |
> |      **Hyena** |           38.4%           |     1.7x    |
> | **Long Convs** |           64.3%           |     2.4x    |
> |    **SaShiMi** |           22.1%           |     1.3x    |
> |   **HyenaDNA** |           81.8%           |     4.4x    |
>
> As you can see, SaShiMi spends the least amount of time computing the convolution, and thus has the least amount of speedup. HyenaDNA spends the most amount of time in the convolution, and has the most speedup. We have added this table to Appendix C.
>
> **Results on Small and Medium Sequence Tasks**
> M2-BERT is a small-sequence task (sequence length 128, benchmarked in Table 4 of the original submission). We also provide benchmarks of Hyena-2.7B at different sequence lengths (2K-16K), and Hyena-s-4K, which are both medium-sequence tasks.
>
> **Applications Outside Machine Learning**
> The FFT is very widely used outside of machine learning, so we look forward to applications outside of machine learning as exciting future work for this project. We have chosen to focus on machine learning applications for this paper, since ICLR is a machine learning conference. However, we note that S4/SaShiMi and Hyena, which we benchmark in the paper, are built on signal processing primitives. We also evaluate on HyenaDNA, a scientific application, and we are sure there are many others that may be relevant.
>
> **Limitations and Extensions of the Method**
> For now, FlashFFTConv is limited to sequence lengths up to 4M. We look forward to further extending these sequence lengths, possibly to billions. This will require further innovation, since we will encounter GPU HBM limitations.
>
> **Potential Negative Impact**
> For FFT convolutions, FlashFFTConv is faster across all sequence lengths. However, FlashFFTConv does currently not support fp32 precision, which could be necessary for some applications with high-precision requirements. Some FFT applications in science may require double precision, which FlashFFTConv currently does not support.

---

> > ### Comment · Reviewer_8BY8 · 2023-11-23
> > **Thanks for the response.**
> >
> > I really appreciate the authors' response. I raised my score from 5 to 6.

---

### Official Review · Reviewer_Bh63 · 2023-11-01

**Soundness:** 3 good
**Presentation:** 3 good
**Contribution:** 3 good
**Rating:** 8
**Confidence:** 3

**Summary:**

In this paper, the authors propose an efficient method to compute the convolution of long sequences by exploiting new computational features available on modern GPUs, tensor cores. The motivation for this study comes from the long sequences that are typically encountered in language and time-series models that require large filters, in stark contrast with the usually small filters utilized in convolutional vision models. The method is based on a matrix interpretation of the FFT algorithm, specifically a p-Monarch formulation, that decomposes the FFT operations into a small set of multiplications that are performed using the much higher arithmetical intensity afforded by tensor cores on modern A100 and H100 GPUs. The authors discuss several algorithmic details required to achieve high performance using this FFT expression, such as parallelizing over the sequence, instead of a direct distribution of work over the batch and hidden dimension and exploiting several properties related to the nature of the FFTs of interest to large sequence models, such as folding a K element real-to-real transform into a K/2 complex transform. With these observations, the authors were able to significantly extend the applicability of the proposed method to sequences up to 32K 16-bit entries processed within a single threadblock. Performance studies on long sequences demonstrate the performance improvements achieved compared to the baseline FFT implementation available in PyTorch and a fused version implemented using the cuFFTdx library.

**Strengths:**

- Generally well-written with an extensive presentation of the algorithmic and system-related details provided in the appendices.
- Highlights a key difference between traditional convolution filter sizes present in vision models versus larger models and utilizes a decomposition that takes into account modern architectural features available on GPUs, tensor cores specifically.
- The proposed strategy to process the inputs over sequences effectively maximizes the number of elements that may be processed in a single thread block and maximally utilizes the tensor cores with more work per block.
- By fusing multiple operations into a single kernel the authors increase the arithmetic intensity of the conv kernel to move it away from the memory boundness of a single FFT call to a more compute-rich fused counterpart.

**Weaknesses:**

Major:
- I would appreciate a clearer distinction between the order-p Monarch FFT and the classic (Bailey) 4-step FFT. The 4-step FFT may also be computed using the Fourier matrices in $\mathcal{O}(n^2)$ work but, as noted in the text, this expense is reduced substantially by realizing a DFT will perform the same action in $\mathcal{O}(n \log n)$ work. It's not clear from the description how the reader to reconcile these 2 competing ideas regarding the reduction in computational work and improvement in overall performance. I suppose the idea is that although the algorithmic expense of the matrix formulation is higher the small sizes of the blocked inputs coupled with the increased throughput of the tensor core units vs the general arithmetic path makes the overall algorithm performance profitable, is that the correct way to think about this issue?

Minor:
- HBM in section 2.2 introduced as global memory instead of high-bandwidth memory
- I'm not convinced Figure 2 adds any meaningful insight into the differences between the different broadcasting strategies. Parallelization over the batch and hidden dimensions seems clear but the alternate strategy to parallelize over sequences is less informative.
- $\sigma_H$ and $\sigma_S$ are defined in section 3.3 but never referenced in the cost model or definition of $w(i)$.
- The components of the cost model could use a bit more explanation to ensure the reader is aware of the origin of each component and its relationship to the algorithmic definition.
- Is the precision of the datatypes ever mentioned in the main text? I see it is referenced in Appendix D as a 16-bit type.

**Questions:**

- Are the arrows in Figure 3 in the correct positions? The text references tradeoff or crossover points between the different order-p decompositions being of interest but 2 of the arrows seem to reference downward slopes. Maybe I'm interpreting either the text or the graph incorrectly.
- Does the cufftDx variation use also fold the inputs to perform an order N/2 complex transform instead of an order N real-to-real transform?

---

> ### Author Response · Authors · 2023-11-16
> **Response to Reviewer Bh63**
>
> We thank the reviewer for the positive feedback. We answer the questions below:
>
> **Differences between Monarch, Bailey, and 4-step FFT.** Mathematically, all these algorithms are equivalent to each other. For a more in-depth discussion, Van Loan [1] is a great resource on how different FFT algorithms are almost always mathematically equivalent, but use different decomposition strategies for computational reasons.
>
> Historically, Bailey’s FFT algorithm has focused on the **data movement** aspects of the FFT computation – it was designed for settings where the entire input cannot fit into the local memory of a machine. In this way, it is similar to a streaming multiprocessor on a GPU; a single streaming multiprocessor has a limited amount of SRAM available to store the input, so longer sequences require communication to HBM.
>
> The major distinction in the Monarch decomposition is the emphasis on the **compute units** used to compute the individual FFT operations. In the Monarch decomposition, the individual FFT operations are computed using a dense matrix multiply operation. As the reviewer notes, this incurs higher absolute FLOP cost than an $O(N log N)$ algorithm – but the increased throughput of the tensor core units makes up for it.
>
> In FlashFFTConv, we are careful to balance the sizes of the matrix multiply operations with the tensor core units available; we usually only compute 16x16 matrix multiplies, or 32x32 (the tensor cores are 16x16-sized). Larger matrices incur too much additional FLOP cost to be worth it - this is why the blue line in the cost model in figure 3 rises above the rest.
>
> **Sequence Length vs. Batch/Head Parallelism.**
> Thank you for the feedback on Figure 2. Our main point is that we can express multiple parallel FFT operations as a single matrix-matrix multiply operation. We have updated Figure 2, and we welcome additional feedback about its clarity.
>
> **Cost Model**
> Thank you for the feedback on the clarity. We have added additional exposition around the components of the cost model. $\omega(i)$ is just a helper function that lets us simplify the expression a little by not needing to separate out HBM and SRAM intermediates into different functions.
>
> $\omega(i)$ returns $\sigma_H$ if the intermediates of step $i$ of the decomposition need to be stored in HBM, and $\sigma_S$ if they can be stored in SRAM.
>
> **Precision of Datatypes**
> Please see the common response for a discussion about precision of the method.
>
> **Arrows in Figure 3**
> These arrows refer to the “humps” for $p=3$ and $p=4$ for short sequence lengths – cost is high even though overall FLOP cost is low, since those FLOPs do not fully utilize tensor cores. We have updated the figure in the draft to try to make this more clear, and welcome further feedback.
>
> **cufftDx Variant**
> Yes, the cufftDx variant also uses the N/2 trick for the real-to-real transform.
>
> [1] Charles Van Loan, Computational Frameworks for the Fast Fourier Transform. 1992.

---

> > ### Comment · Reviewer_Bh63 · 2023-11-23
> >
> > I thank the authors for their detailed responses that satisfied my initial inquiries. Given their response to my questions and other reviewers I will keep my current rating the same.
> >
> > I have one additional suggestion regarding the discussion of the use of tensor cores. The tensor core API does not support complex datatypes but you are folding the N-length real-valued input into a N/2-length complex-valued input, is the impact of this data transformation reflected anywhere in the description of the implementation? I'm curious if it's as simple as invoking the wmma operation twice on the real and complex portions of the inputs.

---

### Official Review · Reviewer_pq6T · 2023-11-04

**Soundness:** 4 excellent
**Presentation:** 3 good
**Contribution:** 4 excellent
**Rating:** 8
**Confidence:** 4

**Summary:**

This paper proposed to improve the Fast Fourier Transform in the long convolution operation by better utilizing hardware advances. Concretely, it first decoupled the matrix to smaller components such that they can be computed via matrix multiply units (Tensor Cores) in hardwares. Second, through monarch decomposition, it allows the proposed algorithm to process much longer sequences under the constrains of shared memory of GPUs.

Experiments on Hyena and M2-Bert demonstrate that the proposed algorithm achieved significant efficiency, while does not hurt model performance.

**Strengths:**

1. This paper aims to solve an import problem in long-sequence convolution computation: poor hardware utilization for FFT.

2. The proposed algorithm achieved significant efficiency, while maintaining model accuracy/performance.

**Weaknesses:**

NA.

**Questions:**

One question is about the precision of the proposed algorithm: since all the intermediate computations are performed using bf16/fp16, I was wondering how precise they are when comparing with the vanilla FFT implementation under fp32.

---

> ### Author Response · Authors · 2023-11-16
> **Response to Reviewer pq6T**
>
> We thank the reviewer for the positive feedback. For the question about the precision of the method, please refer to the table in the common response.

---

### Author Response · Authors · 2023-11-16
**Common Response (1/2)**

We thank all the reviewers for their time and effort reviewing our paper and for their constructive comments, which have made our paper stronger. We appreciate the positive feedback that our work solves an “important problem” (reviewer pq6T) with a “solid” method and “convincing” experiments (reviewer 8BY8). In this common response, we address some common questions that were brought up by multiple reviewers: a summary of why we need a new algorithm for FFT convolution now (even though FFT has been around for decades) and the precision of the inputs & numerical error.

We’re also excited to report that FlashFFTConv has already started to see some adoption to train new models since our initial submission – it is being used to train long-context retrieval models, and to scale up hybrid gated convolution-attention architectures that rival the best open-source LLMs in quality. We're excited by this rapid adoption and impact, and **appreciate your help in making this work better and even more impactful!**

# Why do we need to optimize the FFT Convolution?
A question brought up by multiple reviewers (Bh63, 8BY8) was why we need to be developing new algorithms for the FFT convolution now, when the FFT has existed in different guises for half a century, and when there have been so many algorithms for the FFT in the past.

The key is that we have a **new use case, with new hardware and bottlenecks.** In our case, a recent wave of new ML architectures such as S4, Hyena, BiGS, CKConv, M2, and more [1-5] require optimization of the FFT convolution, and the characteristics of ML accelerators such as GPUs introduce new bottlenecks around I/O and utilizing matrix-multiply units.

As an aside – developing new FFT algorithms for new bottlenecks has actually been common in the history of the FFT – the “FFT algorithm” is not actually one algorithm, but many different algorithms. These algorithms are all mathematically-equivalent to each other, but they are computationally different depending on the compute and memory characteristics of the systems of the day. For instance, see:
* Winograd FFT [6] to minimize multiplication vs. addition (for old hardware platforms where multiplication was more expensive than addition)
* Bailey’s FFT [7] was originally designed to distribute extremely long FFTs that did not fit in the local memory of a single machine – and is primarily focused with the memory access pattern.
* Hexagonal FFT [8] for data sampled on hexagonal grids

FlashFFTConv is another step in this line of new FFT algorithms designed for new hardware bottlenecks. There are two important points:
* The convolution requires FFT, pointwise multiply, and inverse FFT. This has a different **memory-compute tradeoff** than a single FFT operation.
  * Indeed, a **single FFT** does not have this tradeoff, even on GPUs. On GPU, the compute is fast enough (even without tensor cores) that a single FFT is completely memory-bound – so it doesn’t matter if it’s computed with tensor cores or with general arithmetic operations. See below, for a single FFT of length 1K has similar runtime to torch.clone on H100 (and is actually slightly faster, which may indicate slightly better implementation than torch.clone):

|   **Operation** | **Runtime (ms), B = 64, H = 768** |
|----------------:|:---------------------------------:|
| **torch.clone** |                0.31               |
|         **FFT** |                0.27               |
| **Fused FFTConv (no tensor cores)** | 0.67          |

* When multiple steps of the FFT are fused together, then the memory-compute tradeoff changes, and a **fused FFT convolution is no longer memory-bound** (see the third row above, where a fused FFT convolution takes more time than just reading & writing the output).
  * That is when it is important to use the tensor cores – compute becomes the bottleneck.

That memory-compute tradeoff is why the method is named FlashFFTConv, and not FlashFFT – we specifically focus on the tradeoffs for FFT convolutions on GPU. This is also the **primary difference** between the FlashFFTConv algorithm, and previous statements, like Bailey’s FFT decomposition or the four-step FFT decomposition. The core arithmetic is the same, but previous algorithms emphasize different bottlenecks (i.e., Bailey’s only focuses on the limits of local memory, not the compute units to compute the FFT).

Van Loan [9] gives a good overview of frameworks for computing the FFT, and how they are all mathematically the same formulation, but with different compute characteristics, for those interested.

---

> ### Author Response · Authors · 2023-11-16
> **Common Response (2/2)**
>
> # Precision of FlashFFTConv
> Reviewers pq6T and Bh63 ask about the numerical precision of the method. We currently have implementations in fp16 and bf16.
>
> We present some results on the numerical precision of the method, compared to an fp32 PyTorch implementation of the FFT convolution, for a sample of sequence lengths. We present $L_\inf$ error, the maximum absolute error across the entire output:
>
> |          | **256** | **1K** | **4K** | **32K** | **1M** |
> |---------:|:-------:|:------:|:------:|:-------:|:------:|
> | **fp16** |  3.8e-6 | 7.6e-6 | 1.5e-5 |  6.5e-5 | 4.0e-4 |
> | **bf16** |  3.1e-5 | 3.1e-5 | 6.1e-5 |  6.1e-5 | 6.1e-5 |
>
> We have added these results to the appendix. We plan to support fp32 in the future, but prioritized fp16 and bf16 initially since speed-sensitive training workloads generally train in half-precision (to make the MLPs faster).
>
> [1] Efficiently Modeling Long Sequences with Structured State Spaces. https://arxiv.org/abs/2111.00396
>
> [2] Hyena Hierarchy: Towards Larger Convolutional Language Models. https://arxiv.org/abs/2302.10866
>
> [3] Pretraining Without Attention. https://arxiv.org/abs/2212.10544
>
> [4] CKConv: Continuous Kernel Convolution For Sequential Data. https://arxiv.org/abs/2102.02611
>
> [5] Monarch Mixer: A Simple Sub-Quadratic GEMM-Based Architecture. https://arxiv.org/abs/2310.12109
>
> [6] S. Winograd. On computing the discrete Fourier transform" Math. Comp., 32 (1978) pp. 175–199.
>
> [7] Bailey’s FFT Algorithm. https://en.wikipedia.org/wiki/Bailey%27s_FFT_algorithm
>
> [8] Hexagonal fast Fourier Transform. https://en.wikipedia.org/wiki/Hexagonal_fast_Fourier_transform
>
> [9] Charles Van Loan, Computational Frameworks for the Fast Fourier Transform. 1992.

---

### Public Comment · ~Wenxuan_Tan1 · 2025-05-30
**Potential typo in equation (1)**

It seems that equation (1) is not convolving over the input--the location i is fixed.
It should be $(u \cdot k)[i] = \sum_{j=0}^{N} u[i - j] \cdot k[j]$
?

---

> ### Public Comment · ~Daniel_Fu1 · 2025-05-30
> **Response**
>
> Oh, good catch I think you’re right - we must have switched indexing at some point and missed it. We’ll fix it in the arxiv. Thanks for the comment!

---

### Meta-Review · Area_Chair_MUjn · 2023-12-10

**Metareview:**

The paper proposes a new algorithm for computing convolutions via FFT, specifically useful for long-sequence tasks, more effectively on recent GPUs leveraging tensor core hardware.

Reviewers were consistently positive about the paper, and liked the overall contributions made. This includes convincing experimental results, sufficient novelty and a clearly written paper, overall well-deserving acceptance.

We hope the authors will incorporate the several minor points mentioned by the reviewers during the discussions.

**Justification For Why Not Higher Score:**

Overall convolution based method are still less in the spotlight compared to transformers, even for long range tasks, but it remains to be seen if this paper can slightly move the needle on this

**Justification For Why Not Lower Score:**

All reviewers recommend accept

---

### Decision · Program_Chairs · 2024-01-16

Accept (poster)